# Integration of metabolomics and chemometrics with in-silico and in-vitro approaches to unravel SARS-Cov-2 inhibitors from South African plants

Karabo Maselepe Makoana[1], Clarissa Marcelle Naidoo[1], Muhammad Sulaiman Zubair[2], Mmei Cheryl Motshudi[1], Nqobile Monate Mkolo[1]*

**1** Department of Biology, School of Science and Technology, Sefako Makgatho Health Science University, Pretoria, South Africa, **2** Department of Pharmacy, University of Tadulako, Palu, Indonesia

☉ These authors contributed equally to this work.
* nqobile.mkolo@smu.ac.za

## Abstract

Coronavirus disease (COVID-19) is still a severe concern, especially in Africa with suboptimal intention rates of vaccination. This flagged the requirement of plant-based remedies as an alternative treatment. In this study we integrated metabolomics and chemometrics approaches with In silico and In vitro approaches to accelerate and unravel compounds from commonly used South African plants that may inhibit SARS-CoV-2 main protease. The selected commonly used plants, *Artemisia afra* and *Artemisia annua*, were found to be non-toxic against Vero cells, as determined by the resazurin cell viability assay. Metabolites profiling revealed eighty-one compounds and the top three hit compounds, quercetin 3-O-(6"-acetyl-glucoside), 2"-O-acetylrutin, and quercetin 3-(6"-malonyl-glucoside), had binding affinities of -9.3 kcal/mol, -9.5 kcal/mol, and -9.3 kcal/mol, respectively. The 2"-O-acetyl group of the rutin moiety and quercetin moiety produces a hydrogen bond with the amide nitrogen of His41 and with the side chain carboxylate of Cys145, respectively. Molecular dynamics simulations revealed a stable binding of the docked complexes. In silico observations were validated by In vitro bioassay, which flagged the ability of these compounds to inhibit SARS-CoV-2 3CLpro. The collected analysed data of this study does not only draw special attention to the surfaced 2"-O-acetylrutin as the best suitable inhibitor of SARS-CoV-2 3CLpro, but also indirectly reveals the importance of integrating metabolomics and chemometrics approaches with In silico and In vitro approaches to accelerate and unravel compounds from South African commonly used plants.

## Introduction

Severe acute respiratory syndrome coronavirus 2 (SARS-CoV-2), the causative virus for coronavirus disease (COVID-19), previously referred to as the "2019 novel coronavirus" or "2019-nCoV," is a contagious illness resulting from a newly identified coronavirus [1]. The outbreak of this disease was first recognized in Wuhan Hubei, China in December 2019 by the World

**Data availability statement:** All relevant data are within the manuscript and its Supporting Information files

**Funding:** South African Medical Research Council, grant number MKOLO24/25. The funders had no role in study design, data collection and analysis, decision to publish, or preparation of the manuscript.

**Competing interests:** The authors have declared that no competing interests exist.

Health Organization [2]. As of May 2024, SARS-CoV-2 has infected over 700 million people globally and caused more than 6.8 million deaths worldwide [3,4]. While the virus primarily affects the lower respiratory tract and can cause severe, life-threatening respiratory disease, it can also impact other organs such as the liver, gastrointestinal system, kidneys, heart, and central nervous system, resulting in the occurrence of multiple organ failure [5]. Like other respiratory viruses, SARS-CoV-2 is highly contagious and is primarily transmitted through the respiratory tract [6]. The virus spreads from person to person through an infected individual's nasal, mouth, eye, mucosal secretions and through direct inhalation of droplets released when the person coughs or sneezes [7,8].

Despite the development of several SARS-CoV-2 vaccines, the projected expectations in the year 2020 and 2021 concerning the advancement of SARS-CoV-2 vaccinations has not materialized, notably in impoverished countries [9–11]. In South Africa approximately 41% of the population is vaccinated with one dose of vaccine [10,11,12]. Uncertainty persists due to concerns over potential side effects, distrust of health authorities, and the spread of misinformation [13,14]. Furthermore, the emergence of new SARS-CoV-2 variants challenges the efficacy of existing vaccines [13,15,16]. Moreover, long-term effects of Covid (Long COVID), still significantly impact quality of life and productivity, underscoring the need for targeted treatments for acute infections and chronic symptoms [17–19]. The exploration of protease targeting has surfaced as a hopeful direction in the search for antiviral drugs, evident in the effectiveness of protease inhibitors against Human immunodeficiency virus (HIV) and hepatitis C virus (HCV) [20]. A recent 2020 study by Gordon et al, [21], detected 332 interactions between human proteins and SARS-CoV-2, establishing protein-protein associates, with the 3C like protease (3CLPro), or main protease (Mpro) of SARS-CoV-2 being the target protein of interest. The 3CLPro is a 33.8 kDa cysteine protease that plays an important function in the life cycle of the virus by processing the polyprotein at up to 11 conserved sites, resulting in the release of most non-structural protein of SARS-CoV 2, it is an attractive target for identifying anti-SARS-CoV 2 drugs because of its central role and lack of homologous proteins in human cells [22].

Thus, we investigated 3CLPro of SARS-CoV-2 as a target for mining compounds of South African plants. The use of South African plants is prompted by a multitude of data concerning commonly used South African medicinal plants and their promising pharmacological advantages. To date, researchers have not provided attention towards exploring potential compounds associated with commonly used South African medicinal plants as potential inhibitors for different targets of SARS-CoV-2, including 3CLpro. The selected commonly used plants are *Artemisia afra* (African wormwood) and *Artemisia annua* (Sweet Annie), known for their traditional uses in managing viral infections, including SARS-CoV-2 infection [23–26]. These plants belong to the biggest genus in the Asteraceae family [23]. *A. afra* is a fragrant perennial shrub with multiple stems, found throughout the mountainous areas of Africa, from Ethiopia to South Africa [26,27]. The plant has a long history of traditional uses in treating various health conditions such as coughs, colds, headaches, chills, asthma, malaria, diabetes, convulsions, and fever [23,28]. *A. annua* is a prevalent species of wormwood originally from temperate Asia but now established in various countries, including South Africa, well-known for its high concentration of artemisinin a powerful anti-malarial compound belonging to the sesquiterpene lactone group [23,29,30]. Despite these plants sharing molecular similarities, *A. afra* has far lower levels of artemisinin compared to *A. annua*, but has higher concentrations of monoterpenes, sesquiterpenes, and polyphenols, which are responsible for the distinct medicinal properties exhibited by the plant [23,31].

Herein we attempted to integrate metabolomics and chemometrics approaches with In silico and In vitro approaches to accelerate and unravel South African plant-based compounds that can inhibit SARS-CoV-2 main protease.

## Materials and methods

### Ethical approval and preparation of plant material

The Sefako Makgatho Health Sciences University Research and Ethics Committee approved the study (Ethics reference no: SMUREC/S/62/2024:PG). South African Department of Agriculture and Rural Development-Nature Conservation granted the plant collection permit to Prof Mkolo Nqobile Monate (Permit No. CF6-0234). Afterward, the leaves of *A. afra* and *A. annua* were collected from Hartbeespoort, South Africa (25.7236◦ S, 27.9653◦ E) in February 2023. The *A. afra* and *A. annua* specimens were authenticated by Curator of South African National Herbarium and were allocated voucher No. 903 and No. 902, respectively.

The collected plant samples were powdered in an electric grinder (Krups Burr Grinder). Subsequently, 5 g of each powder samples was extracted using 50 mL of methanol (Merck, Johannesburg, South Africa). The samples were shaken for 30 min on a Labotec shaker and then centrifuged for about 15 min at 3000 revolutions per minute. The debris was filtered using Whatman No. 1 filter paper and the methanolic crude extracts were dried at room temperature. Stock solutions prepared from these crude extracts were utilized for In vitro biological assessment.

### Determination of cell viability

The resazurin cell viability assay was conducted using Vero cell lines, purchased from Cellonex Separation Scientific, Johannesburg, South Africa. The cells were maintained in Dulbecco's Modified Eagle's Medium (DMEM) supplemented with 10% fetal bovine serum (FBS) (Thermo Scientific, Waltham, MA, USA). Vero cells ($1.25 \times 10^6$ cells/mL) were cultivated at 37°C in a humidified 5% $CO_2$ atmosphere incubator for 24 h [32]. In a sterile laminar flow, the cells were seeded at 5000 cells/well density in 100 µL media. The cells were then treated with stock solutions (10 mg/mL) of *A. afra* and *A. annua* dried methanolic extracts which were dissolved in 100% DMSO, then diluted further with DMEM to attain 250, 125, 62.5, 31.25, 15.63, and 7.81 µg/mL concentrations. Subsequently, 20 µL of resazurin dye (Sigma TOX-8) was added. The reduction levels of resorufin fluorescence were quantified and monitored at 590 nm wavelength (EM), with an excitation wavelength (EX) observed at 560 nm using Modulus II Multifunction Plate Reader (Turner Biosystems, Sunnyvale, CA, USA). Concurrently, quantification of absorbance readings and resazurin peak absorbance were observed at 570 nm and 600 nm wavelengths, respectively.

### Metabolomics approach for tentative identification and quantification of bioactive compounds

**Chemicals and reagents.** Reagents that were utilized are methanol (Merck, Rahway, NJ, USA), Acetonitrile (Merck, Rahway, NJ, USA) and DL-o-Chlorophenylalanine (Merck, Rahway, NJ, USA) and formic acid (Merck, Rahway, NJ, USA).

**Preparation of samples.** The initial procedure of sample preparation included lyophization of *A. annua* and *A. afra* leaves samples. Each dried, and ground powdered sample of 50 mg was transferred into a 5 mL homogenizing tube, mixed at 30 Hz through the use of a MM 400 mixer with the aid of four metal balls (5 mm). Subsequently, 800 µL of 80% methanol was added into each sample, and the mixed solution was vortexed for 30 s and sonicated for 30 min at 4°C. After 1 h of samples storage at -20°C, each sample was centrifuged for 5 min at 12,000 rpm, 4°C. The vial mixture of 200 µL of each supernatant and 5 µL of DL-o-Chlorophenylalanine (140 µg/mL) was further utilized for liquid chromatography–mass spectroscopy (LC-MS) analysis. The quality control (QC) samples were prepared with an

equal addition from each sample to evaluate the methodology and the stability of the LC-MS system.

The same amount of extract was spiked into every sample, as well as used as quality control samples (QC samples). The QC sample is used to indicate the stability of the LC-MS system. Compound Discover (3.0, Thermo) was used to import the total ion current in UPLC-MS/MS raw data acquired, aligned through *m/z* value and the retention time of ion signals. Relative standard deviation (RSD) was calculated using the ion features of QC samples. Distribution of the RSD (%) was less than 30%, which implies that the analysis method was robust. Thus, the data was further utilized for subsequent mentioned analysis.

**Metabolites profiling using UPLC-MS/MS.** The chromatographic separation was performed by Ultimate 3000LC combined with Q Exactive MS (Thermo, Waltham, MA, USA), and screened with ESI-MS [33]. The LC system consist of an ACQUITY UPLC HSS T3 (100 × 2.1 mm × 1.8 μm) with Ultimate 3000LC. The mobile phase consisted of 0.05% formic acid in water (A) and acetonitrile (B) with gradient elution conditions of 0–1 min, 95% A; 1–12 min, 95%–5% A; 12–13.5 min, 5% A; 13.5–13.6 min, 5–95% A; 13.6–16 min, 95% A. The mobile phase flow rate was 0.3 mL·min − 1. The column temperature was sustained at 40 °C, and the sample was maintained at 4 °C. The following conditions were utilized for mass spectrometry parameters in electrospray ionization ESI + modes: heater temperature of 300 °C; auxiliary gas flow rate, 15 arb; sheath gas flow rate, 45 arb; sweep gas flow rate, 1 arb; spray voltage, 3.0 kV; S-Lens RF level, 30%; capillary temperature of 350 °C.

**Ligand binding pocket prediction.** A crucial step in the drug discovery process is identifying protein-ligand binding sites. The 3-D structure of SARS-CoV-2 3CLpro (PDB code: 6M2N) was analyzed to identify the binding sites for the selected compounds. The binding site was predicted using a template free machine learning algorithm template of P2Rank which is integrated in PrankWeb [34]. This functions by means of incorporating local chemical neighborhood ligand capability at junctures situated on a protein surface that is accessible to solvents. The ligand binding sites generated are created by clustering junctures or points with greater ligand capability scores [34].

**Molecular docking.** The optimal conformation, orientation and binding affinity (kcal/mol) of the identified 81 compounds (Table 1), through metabolite profiling were docked against SARS-CoV-2 3CLpro (PDB code: 6M2N). The chemical structures of these compounds were first constructed with ChemDraw 21.0, and subsequently optimized with Chem3D, utilizing Merck Molecular Force Field (MMFF94). Simultaneously, non-receptor heteroatoms such as water and ions were removed for the preparation of SARS-CoV-2 protein targets. Protein atoms were also afforded with Kollman charges. In relation to proteins with native ligands, the grid boxes were arranged according to their location. AutoDock Vina 1.2.3 version 2021 was utilized for molecular docking simulation which was created according to the grid dimensions of the protein targets. The dimensions for the 3C-like protease (3CLpro) (PDB code: 6M2N) were set at 30 Å x 30 Å x 30 Å (x = -32.981, y = -65.135, z = 41.719), in terms of protein with absent native ligand, we used Probability score parameters of PrankWeb's binding site predictions.

**Molecular dynamics simulation.** In this study, we conducted molecular dynamics simulation on the SARS-CoV-2 3CLpro protein. Simulations were conducted to assess the flexibility and stability of the selected three compounds of Quercetin 3-O-(6"-acetyl-glucoside), 2"-O-acetylrutin, and quercetin 3-(6"-malonyl-glucoside) which were docked. The selected three compounds were based on their high binding affinity capabilities (Table 1). Molecular dynamics (MD) simulations were conducted using the Amber20 package on the three compounds, engaging the AMBER FF99SB force field for protein structure preparation. The Antechamber module in AmberTools20 was utilized to create topology arrangement files

**Table 1. Identified and docked compounds from *A. annua* and *A. afra* (ESI + scan).**

| No. | Compound name | RT [min] | m/z | Formula | Monoiso-topic Mass | Delta (ppm) | Log2 (FC) | T-Test | Log$^{10}$ (p value) | VIP | Binding affinity to 3CLPro (Kcal/mol) |
|---|---|---|---|---|---|---|---|---|---|---|---|
| 1 | Fomepizole | 2,75 | 83,06094 | C4H6N2 | 82,053101 | 7 | 3,50 | 1,59E-09 | 8,80 | 1,52 | -3.6 |
| 2 | Piperidine | 1,227 | 86,09691 | C5H11N | 85,08914936 | 6 | 4,59 | 2,02E-06 | 5,69 | 1,73 | -3.3 |
| 3 | Methylpyrazine | 2,893 | 95,06079 | C5H6N2 | 94,0530982 | 4 | 5,70 | 3,46E-06 | 5,46 | 1,94 | -3.7 |
| 4 | 5-Methyl-2-furancarboxaldehyde | 1,024 | 111,04423 | C6H6O2 | 110,0367794 | 2 | -3,99 | 7,98E-07 | 6,10 | 1,62 | -4 |
| 5 | 1,3,4-Oxadiazepine | 2,644 | 138,06601 | C4H4N2O | 96,032364 | 1 | 10,04 | 7,82E-08 | 7,11 | 2,56 | -3.7 |
| 6 | 5-Ethyl-2-methylpyridine | 2,804 | 139,12275 | C8H11N | 121,0891494 | 2 | -3,98 | 8,79E-03 | 2,06 | 1,52 | -4.2 |
| 7 | Methyl 2-thiofuroate | 0,897 | 143,0188 | C6H6O2S | 142,0088501 | 19 | 4,34 | 8,94E-08 | 7,05 | 1,74 | -3.8 |
| 8 | 3-Amino-2-cyclohexenone | 4,936 | 150,03087 | C6H9NO | 111,068413 | 6 | -4,06 | 9,84E-07 | 6,01 | 1,63 | -4.5 |
| 9 | Parvoline | 3,933 | 153,13835 | C9H13N | 135,104797 | 2 | 3,96 | 1,36E-05 | 4,87 | 1,65 | -4.5 |
| 10 | 3-Amino-2,2-dimethylpropanoic acid | 0,92 | 156,04193 | C5H11NO2 | 117,0789786 | 1 | 3,63 | 7,72E-06 | 5,11 | 1,54 | -4.2 |
| 11 | Propylpyrazine | 1,311 | 164,11804 | C7H10N2 | 122,0843983 | 1 | 8,05 | 5,30E-06 | 5,28 | 2,29 | -4.1 |
| 12 | Oxoglutaric acid | 5,003 | 169,01288 | C5H6O5 | 146,0215233 | 13 | 3,55 | 1,29E-04 | 3,89 | 1,51 | -4.5 |
| 13 | Quinolacetic acid | 2,849 | 169,04939 | C8H8O4 | 168,0422587 | 1 | -3,61 | 7,80E-06 | 5,11 | 1,53 | -5.2 |
| 14 | 4-Phenyl-3(2H)-pyridazinone | 2,954 | 173,07071 | C10H8N2O | 172,06366 | 1 | -4,82 | 4,03E-06 | 5,39 | 1,78 | -5.6 |
| 15 | 4-Guanidino-1-butanol | 0,787 | 173,13956 | C5H13N3O | 131,105865 | 1 | 4,88 | 4,27E-04 | 3,37 | 1,76 | -4 |
| 16 | Nicotyrine | 2,672 | 176,11808 | C10H10N2 | 158,0843983 | 1 | 5,18 | 1,38E-05 | 4,86 | 1,86 | -5 |
| 17 | Anatabine | 2,75 | 178,13366 | C10H12N2 | 160,100052 | 1 | 8,36 | 8,25E-07 | 6,08 | 2,34 | -5.3 |
| 18 | 4-(1H-Pyrazol-1-yl)-1-butanol | 2,644 | 182,12868 | C7H12N2O | 140,094955 | 1 | 7,00 | 1,09E-05 | 4,96 | 2,14 | -4.2 |
| 19 | Choline sulfate | 0,988 | 184,06353 | C5H13NO4S | 183,0565291 | 1 | -3,69 | 1,21E-05 | 4,92 | 1,55 | -4 |
| 20 | 1,4-Octadien-1-ylbenzene | 8,255 | 187,1479 | C14H18 | 186,140854 | 1 | 3,59 | 7,32E-09 | 8,14 | 1,53 | -5.1 |
| 21 | Amino(1H-indol-2-yl) acetic acid | 3,17 | 191,08118 | C10H10N2O2 | 190,074234 | 1 | -4,41 | 5,44E-09 | 8,26 | 1,78 | -5.7 |
| 22 | 5-Allyl-6-methyl-4(1H)-pyrimidinone | 2,32 | 192,11293 | C8H10N2O | 150,079315 | 2 | 3,81 | 6,02E-06 | 5,22 | 1,58 | -4.7 |
| 23 | 6-Amino-2,4,5-trimethyl-3-pyridinol | 2,096 | 194,12868 | C8H12N2O | 152,094955 | 1 | 8,26 | 3,94E-06 | 5,40 | 2,32 | -5 |
| 24 | 2'-Hydroxynicotine | 2,769 | 196,14422 | C10H14N2O | 178,1106131 | 1 | 11,10 | 1,38E-06 | 5,86 | 2,70 | -5.5 |
| 25 | 3,6,8-Dodecatrien-1-ol | 10,71 | 203,14697 | C12H20O | 180,151413 | 12 | 3,60 | 1,02E-07 | 6,99 | 1,60 | -5.1 |
| 26 | 3,6-Dodecadien-1-ol | 6,947 | 205,15849 | C12H22O | 182,1670653 | 11 | 3,77 | 6,44E-08 | 7,19 | 1,57 | -4.5 |
| 27 | 2-Methylenecyclododecanone | 5,774 | 212,20058 | C13H22O | 194,167068 | 1 | 5,85 | 8,57E-06 | 5,07 | 1,96 | -5.1 |
| 28 | 2',3'-Dideoxyuridine | 4,976 | 213,09057 | C9H12N2O4 | 212,0797069 | 17 | -3,49 | 1,25E-07 | 6,90 | 1,54 | -6.1 |
| 29 | 1-Phenylcyclohexanol | 3,774 | 218,15369 | C12H16O | 176,120117 | 1 | 5,05 | 2,56E-07 | 6,59 | 1,82 | -5.4 |
| 30 | 4-Pyridoxic acid | 5,862 | 225,09062 | C8H9NO4 | 183,0531578 | 16 | -5,38 | 5,82E-08 | 7,24 | 1,97 | -4.9 |
| 31 | 3'-Deoxythymidine | 5,839 | 227,1062 | C10H14N2O4 | 226,0953569 | 16 | -5,95 | 7,73E-07 | 6,11 | 2,02 | -6.4 |
| 32 | Pyroglutamylvaline | 5,782 | 229,12191 | C10H16N2O4 | 228,111007 | 16 | -3,47 | 4,94E-06 | 5,31 | 1,50 | -5.5 |
| 33 | 2-Hydroxy-3-phenylcyclohexanone | 3,169 | 232,13287 | C12H14O2 | 190,09938 | 11 | 6,43 | 1,10E-06 | 5,96 | 2,05 | -5.8 |
| 34 | 3'-Amino-3'-deoxythimidine | 4,038 | 242,11701 | C10H15N3O4 | 241,106256 | 14 | -5,09 | 2,30E-06 | 5,64 | 1,85 | -6.4 |
| 35 | Thymidine | 5,406 | 243,10105 | C10H14N2O5 | 242,0902716 | 14 | -6,83 | 2,21E-06 | 5,66 | 2,13 | -6.5 |
| 36 | Germacrone-13-al | 4,513 | 250,17963 | C15H20O2 | 232,1463299 | 2 | -6,74 | 4,10E-06 | 5,39 | 2,10 | -6.3 |
| 37 | 2-Hydroxyacorenone | 9,271 | 259,16643 | C15H24O2 | 236,17763 | 2 | 4,60 | 4,58E-07 | 6,34 | 1,74 | -6.5 |
| 38 | Artemorin | 7,694 | 266,17446 | C15H20O3 | 248,1412445 | 2 | 6,29 | 9,52E-07 | 6,02 | 2,03 | -6.8 |
| 39 | 3alpha-Hydroxyoreadone | 6,425 | 275,12477 | C14H20O4 | 252,1361591 | 2 | 5,16 | 1,95E-06 | 5,71 | 1,84 | -6.9 |
| 40 | 2',3'-Dideoxyadenosine | 6,718 | 277,14047 | C10H13N5O2 | 235,1069247 | 1 | 4,34 | 1,56E-11 | 10,81 | 1,70 | -6.1 |
| 41 | 5-Methyldeoxycytidine | 2,036 | 283,13964 | C10H15N3O4 | 241,106256 | 2 | 4,39 | 4,92E-06 | 5,31 | 1,69 | -6.3 |
| 42 | 3-Hydroxy-2-oxobutyl nonanoate | 6,094 | 286,20067 | C13H24O4 | 244,167465 | 3 | 8,00 | 6,42E-06 | 5,19 | 2,34 | -4.9 |
| 43 | Lactucin | 3,633 | 318,13286 | C15H16O5 | 276,0997736 | 2 | -5,38 | 4,31E-04 | 3,37 | 1,88 | -7 |
| 44 | N2-Galacturonyl-lysine | 6,257 | 323,14815 | C12H22N2O8 | 322,1376157 | 10 | -6,46 | 2,79E-06 | 5,55 | 2,06 | -6 |
| 45 | 7-Methylinosine | 1,031 | 325,139 | C11H15N4O5 | 283,1042446 | 2 | -6,80 | 7,61E-06 | 5,12 | 2,11 | -6.3 |
| 46 | Aflatoxin G2 | 6,708 | 331,08041 | C17H14O7 | 330,0739528 | 2 | -7,62 | 4,45E-06 | 5,35 | 2,23 | -8.4 |

*(Continued)*

**Table 1.** (Continued)

| No. | Compound name | RT [min] | m/z | Formula | Monoiso-topic Mass | Delta (ppm) | Log2 (FC) | T-Test | Log$^{10}$ (p value) | VIP | Binding affinity to 3CLPro (Kcal/mol) |
|---|---|---|---|---|---|---|---|---|---|---|---|
| 47 | Protocatechuic acid 4-glucoside | 2,983 | 339,06788 | C13H16O9 | 316,0794321 | 2 | -5,71 | 1,73E-05 | 4,76 | 1,96 | -6.8 |
| 48 | Glutamyllysine | 2,35 | 339,16556 | C11H21N3O5 | 275,1481208 | 5 | 5,59 | 7,30E-06 | 5,14 | 1,91 | -5.6 |
| 49 | 6-Ketoestriol | 6,526 | 344,18475 | C18H22O4 | 302,1518092 | 3 | -4,86 | 1,91E-05 | 4,72 | 1,78 | -7.5 |
| 50 | Glucitol-lysine | 5,161 | 352,21098 | C12H26N2O7 | 310,1740012 | 9 | 5,03 | 8,79E-09 | 8,06 | 1,83 | -5.7 |
| 51 | Nicotine glucuronide | 1,84 | 356,18075 | C16H22N2O6 | 338,1477864 | 2 | 8,28 | 5,38E-06 | 5,27 | 2,33 | -8 |
| 52 | 4',5,7-Trihydroxy-6-prenylflavanone | 3,227 | 358,16398 | C20H20O5 | 340,1310737 | 3 | -9,01 | 2,18E-06 | 5,66 | 2,43 | -7.4 |
| 53 | 3-Epinobilin | 3,983 | 364,21086 | C20H26O5 | 346,1780239 | 3 | -6,27 | 1,35E-04 | 3,87 | 2,03 | -7.4 |
| 54 | Zeranol | 4,259 | 364,2109 | C18H26O5 | 322,1780239 | 3 | -5,67 | 8,31E-05 | 4,08 | 1,92 | -7.4 |
| 55 | Cibaric acid | 4,971 | 366,22678 | C18H28O5 | 324,193674 | 2 | 4,11 | 5,96E-07 | 6,22 | 1,64 | -5.7 |
| 56 | Isocolumbin | 3,074 | 376,1747 | C20H22O6 | 358,1416384 | 2 | -8,16 | 7,32E-06 | 5,14 | 2,31 | -8.1 |
| 57 | Lactol | 3,939 | 378,19013 | C20H24O6 | 360,1572885 | 3 | -7,67 | 7,07E-07 | 6,15 | 2,24 | -3.6 |
| 58 | Hydroxyisonobilin | 3,359 | 380,20585 | C20H26O6 | 362,1729386 | 2 | -5,12 | 5,18E-05 | 4,29 | 1,82 | -7.9 |
| 59 | 1,2-Anhydridoniveusin | 1,885 | 394,18523 | C20H24O7 | 376,1522031 | 2 | -8,65 | 1,79E-05 | 4,75 | 2,38 | -7.7 |
| 60 | Kasugamycin | 1,915 | 402,14996 | C14H25N3O9 | 379,159088 | 4 | 8,18 | 3,56E-07 | 6,45 | 2,31 | -7.1 |
| 61 | Eupachloroxin | 3,586 | 446,1566 | C20H25ClO8 | 428,12381 | 2 | -7,90 | 2,02E-05 | 4,69 | 2,27 | -7.5 |
| 62 | MG(i-20:0/0:0/0:0) | 14,504 | 409,32811 | C23H46O4 | 386,33961 | 2 | 5,86 | 1,59E-04 | 3,80 | 1,96 | -4.8 |
| 63 | Biocytin | 4,469 | 414,21116 | C16H28N4O4S | 372,1831261 | 14 | -6,26 | 1,38E-03 | 2,86 | 1,99 | -6.1 |
| 64 | Sergliflozin A | 3,22 | 418,18493 | C20H24O7 | 376,1522031 | 3 | -7,79 | 6,77E-07 | 6,17 | 2,26 | -7.7 |
| 65 | N-Acetyl-9-aminominocycline, (4R)- | 4,66 | 420,20061 | C20H31NO7 | 397,2100523 | 3 | -9,55 | 4,18E-06 | 5,38 | 2,50 | -7.6 |
| 66 | Enicoflavine | 3,338 | 423,17518 | C10H13NO4 | 211,084457 | 2 | 3,64 | 2,11E-06 | 5,67 | 1,54 | -5 |
| 67 | Eurycomanol | 1,378 | 428,19056 | C20H26O9 | 410,1576824 | 2 | -5,56 | 1,18E-05 | 4,93 | 1,90 | -7.2 |
| 68 | Aloesol 7-glucoside | 4,058 | 438,17503 | C19H24O9 | 396,1420324 | 2 | 3,43 | 2,49E-07 | 6,60 | 1,51 | -8.4 |
| 69 | 25-Hydroxyvitamin D3-26,23-lactone | 11,375 | 451,28331 | C27H40O4 | 428,2926598 | 3 | 6,77 | 1,30E-06 | 5,89 | 2,13 | -8.2 |
| 70 | Fluocinolone | 2,966 | 454,20623 | C21H26F2O6 | 412,1697449 | 6 | -8,07 | 3,77E-06 | 5,42 | 2,30 | -7.5 |
| 71 | LysoPA(18:0/0:0) | 10,568 | 461,26759 | C21H43O7P | 438,2746407 | 8 | 5,34 | 4,21E-08 | 7,38 | 1,87 | -4.9 |
| 72 | Davallialactone | 5,955 | 465,11703 | C25H20O9 | 464,1107322 | 2 | 4,28 | 1,62E-05 | 4,79 | 1,67 | -8.5 |
| 73 | 6"-O-Acetylglycitin | 5,524 | 489,13817 | C24H24O11 | 488,1318616 | 2 | 4,53 | 1,14E-05 | 4,94 | 1,73 | -8.1 |
| 74 | Quercetin 3-O-(6"-acetyl-glucoside) | 3,718 | 507,11242 | C23H22O13 | 506,1060408 | 2 | 3,46 | 9,13E-07 | 6,04 | 1,50 | -9.3 |
| 75 | Caryatin glucoside | 4,025 | 529,13309 | C24H26O12 | 506,1424263 | 3 | 4,13 | 3,72E-06 | 5,43 | 1,67 | -8.3 |
| 76 | Gluten exorphin B4 | 6,575 | 547,21614 | C24H27N5O9 | 529,1808775 | 3 | -5,58 | 2,07E-08 | 7,68 | 1,93 | -7.7 |
| 77 | Quercetin 3-(6"-malonyl-glucoside) | 4,937 | 551,10198 | C24H22O15 | 550,09587 | 2 | 6,00 | 7,96E-06 | 5,10 | 2,00 | -9.3 |
| 78 | Sesaminol glucoside | 7,259 | 555,14816 | C26H28O12 | 532,1580764 | 2 | -6,32 | 5,81E-07 | 6,24 | 2,03 | -9.3 |
| 79 | 2"-O-Acetylrutin | 8,193 | 691,12656 | C29H32O17 | 652,1639496 | 1 | 4,94 | 2,19E-07 | 6,66 | 1,80 | -9.5 |
| 80 | Linalool (8-hydroxydihydro-) | 4,627 | 692,28835 | C32H42O14 | 650,257456 | 4 | 5,32 | 4,52E-10 | 9,34 | 1,87 | -4.3 |
| 81 | PA(i-16:0/PGE2) | 12,248 | 808,46495 | C39H69O11P | 744,45775 | 11 | -5,37 | 4,23E-05 | 4,37 | 1,87 | *** |

**Key** Compound 81*** cannot be docked due to the silicon "Si" atom in the structure. ▇▇▇ signifies the top three compounds with the lowest binding energy which have best affinities to 3CLPro. (a) denotes retention time (RT), (b) denotes mass-to-charge ratio (m/z) values, (c) denotes fold change (FC) values and (d) denotes variable importance in projection (VIP) values.

for ligands by employing the general amber force field (GAFF). Successively, the complexes were submerged in truncated octahedral boxes comprising TIP3P water molecules, featuring a buffer region of 10 Å. Ions Cl− or Na+ were subsequently added to neutralize the complexes. Amber20's default parameter settings were used for protein residues. The SANDER module in the Amber20 package was utilized to perform energy minimization and simulation. To ensure a more stable initiating configuration for the succeeding simulations, alleviation of

steric conflicts was conducted. The estimated 10,000 cycles of minimization (MAXCYC) were achieved by utilization of the initial 500 steps of steepest descent (NCYC) and 1000 steps of conjugate gradients. Energy minimized system was subjected to equilibration in constant Number of particles, Volume, and Temperature (NVT) and constant Number of particles, Pressure, and Temperature (NPT) phases. After energy minimization, positioning restraint for constant volume (NVT) of 100 picoseconds (ps) were conducted with 10 kcal/mol restraint force at 310 Kelvin (K) temperature. Subsequently, constant pressure (NPT) equilibration for 100 ps was employed with 1 kcal/mol restraint force at 310 K temperature. Then equilibration for constant pressure (NPT) at 310 K temperature was conducted, taking 100 ps under 2 fs time step. The restraint forces were eliminated during this phase, leading water to attain the equilibrium density in the protein. Algorithms from Langevin dynamics were applied to control temperature in cooperation with the equilibration and position restraints processes. Lastly, a 50 ns production run at 310 K with isotropic position scaling (ntp = 1) and constant pressure conditions (NPT ensemble) was conducted at 2 fs time step.

**In vitro enzyme inhibition of SARS-CoV-2 3CLpro.** The SARS-CoV-2 3CL protease, MBP-tagged assay kit was obtained from BPS Bioscience, San Diego, CA, USA. In vitro enzyme inhibition of SARS-CoV-2 3CLpro was conducted as per the manufacturer's instructions (Cat. No. 79955-1). Prior to initiation of the enzyme reaction, pre-incubation of 30 μl diluted 4 ng 3CLpro-MBP tagged enzyme in assay buffer (containing 1 mM DTT) consisting of test inhibitor solutions in 96 well plates were conducted for 30 min at room temperature. Inhibitor test solutions consisted of different concentrations of plant extracts (250, 125, 62.5, 31.25, 15.63, and 7.81 μg/mL) and compounds (1.5,1, 0.75, 0.5, 0.25, and 0.1 μM) prepared using DMSO (0.5%). Precisely, *A. annua* and *A. afra* extracts and compounds of quercetin 3-O-(6"-acetyl-glucoside) (Sigma-Aldrich), 2"-O-acetylrutin (GlpBio, Monclair, NJ, USA), and quercetin 3-(6"-malonyl-glucoside) (Sigma-Aldrich) were utilized. The positive control consisted of DMSO (1%) with 4 ng of enzyme and substrate (50 μM) with a diluent solution. The blank consisted of DMSO (1%) with substrate (50 μM) without enzyme.

The initiation of enzyme reaction was achieved by adding 10 μl (250 μM) 3CL protease substrate into the wells, affording a reaction volume of 50 μl. Incubation was conducted for 18 hrs at room temperature. The fluorescence intensity was measured at an excitation wavelength of 360 nm and at an emission wavelength of 460 nm using Modulus II Multifunction Plate Reader (Turner BioSystems, Sunnyvale, CA, USA). All tests were conducted in triplicate, and percentage inhibition values for plant extracts and compounds were calculated. The results are also represented as $IC_{50} \pm$ SD.

## Data analysis

Statistical analysis was performed using GraphPad Prism (version 6.07; La Jolla, CA, USA). The experiments were carried out in triplicate and repeated three times. The results were expressed as percentage for Vero cell viability (%) assay. A plot comprising of the series of concentrations versus the percentage inhibition was utilized to calculate the extract/compound concentrations that inhibit 50% of SARS-CoV-2 3CLpro ($IC_{50}$). The results were expressed as Mean ± standard error (SE) for In vitro enzyme inhibition of SARS-CoV-2 3CLPro assay.

The raw electrospray ionization (ESI + modes) data generated from UPLC-MS/MS acquired through Compound Discover (3.0, Thermo) were combined and subjected to the SIMCA-P program version 14.1 (Umetrics, Umea, Sweden) and MetaboAnalyst 5.0 (https://www.metaboanalyst.ca) for multivariate analysis. Prior, multivariate analysis, compounds' spectral data were cross-referenced with Human Metabolome Database (HMDB) (https://hmdb.ca/) and ChemSpider (https://www.chemspider.com) databases. An unsupervised

method was employed on the data set using principal component analysis (PCA) model for data visualization and identification of outliers. Supervised methods were utilized on the data set using orthogonal partial least squares discriminant analysis (OPLS-DA) and partial least squares discriminant analysis (PLS-DA) to categorize and visualize the potential bioactive compounds. Univariate approaches were utilized to discover the potential bioactive compounds that gratify the selection criteria of fold change (FC) > 2.0, VIP > 1.5, and $p$ value < 0.05. $R^2$ and $Q^2$ values were evaluated as the best fit model.

## Results

### Resazurin cell viability assay

The evaluation of Vero cells viability was achieved after 24 h of treatment with *A. annua* and *A. afra,* was assessed using different concentrations ranging from 250 µg/mL to 7.81 µg/mL (Fig 1). Absorbance for resorufin was quantified and the viability of Vero cells was found to be greater than 85.05% and 85.85% after treatment with *A. annua* and *A. afra* at various concentrations, respectively. The reduction levels of resorufin fluorescence were quantified and demonstrated a decreased viability values exceeding 58.96% and 62.51%, post-treatment with *A. annua* and *A. afra* at different concentrations, respectively.

### Metabolites profiling: Univariate and multivariate data analyses

The ESI in positive ionization mode determined 225 peaks of phenolic acids, glycosides, flavonoids, hydroxy fatty acids, carboxylic acid, phenylethanoid, triterpenoids and iridoid glycosides (S1 Fig, Supporting information). Supporting information, S1 and S2 Tables represents comprehensive chemical names and other details of the chemical profiled compounds. Moreover, statistical and chemometric analyses was used to define parameters (compound names, mass-to-charge ratio ($m/z$) values, retention time (rt), formulas, monoisotopic masses, Delta (ppm), fold change (FC) values, $p$ values and variable importance in projection (VIP) values) for elucidation of 81 possible signature compounds that may lead to bioactivity of *A. annua* and *A. afra* (Table 1 and Fig 2A). Principal component analysis (PCA) and partial least squares discriminant analysis (PLS-DA) models were utilized for comparability of metabolic variations that lead to reduction of nonspecific effects and prediction of the prospective compounds that can be 3-chymotrypsin like protease (3CLPro) inhibitors in the two plant groups of *A. annua* and *A. afra*. Univariate analysis revealed the metabolites that gratify the selection criteria of

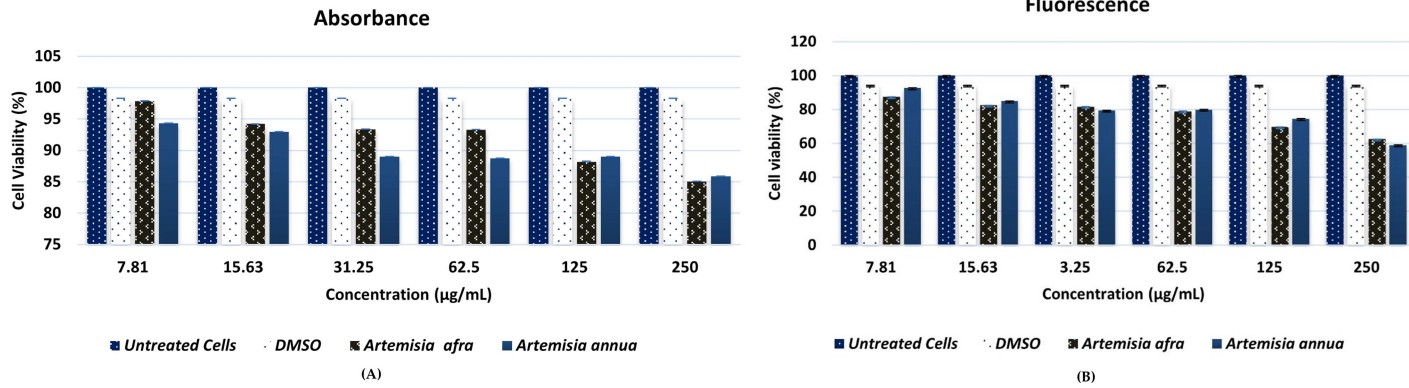

**Fig 1.** Vero cells viabilities post treatments with *A. annua* and *A. afra* at different concentrations (µg/mL); (A) Resorufin absorbance quantification. (B) Resorufin fluorescence quantification.

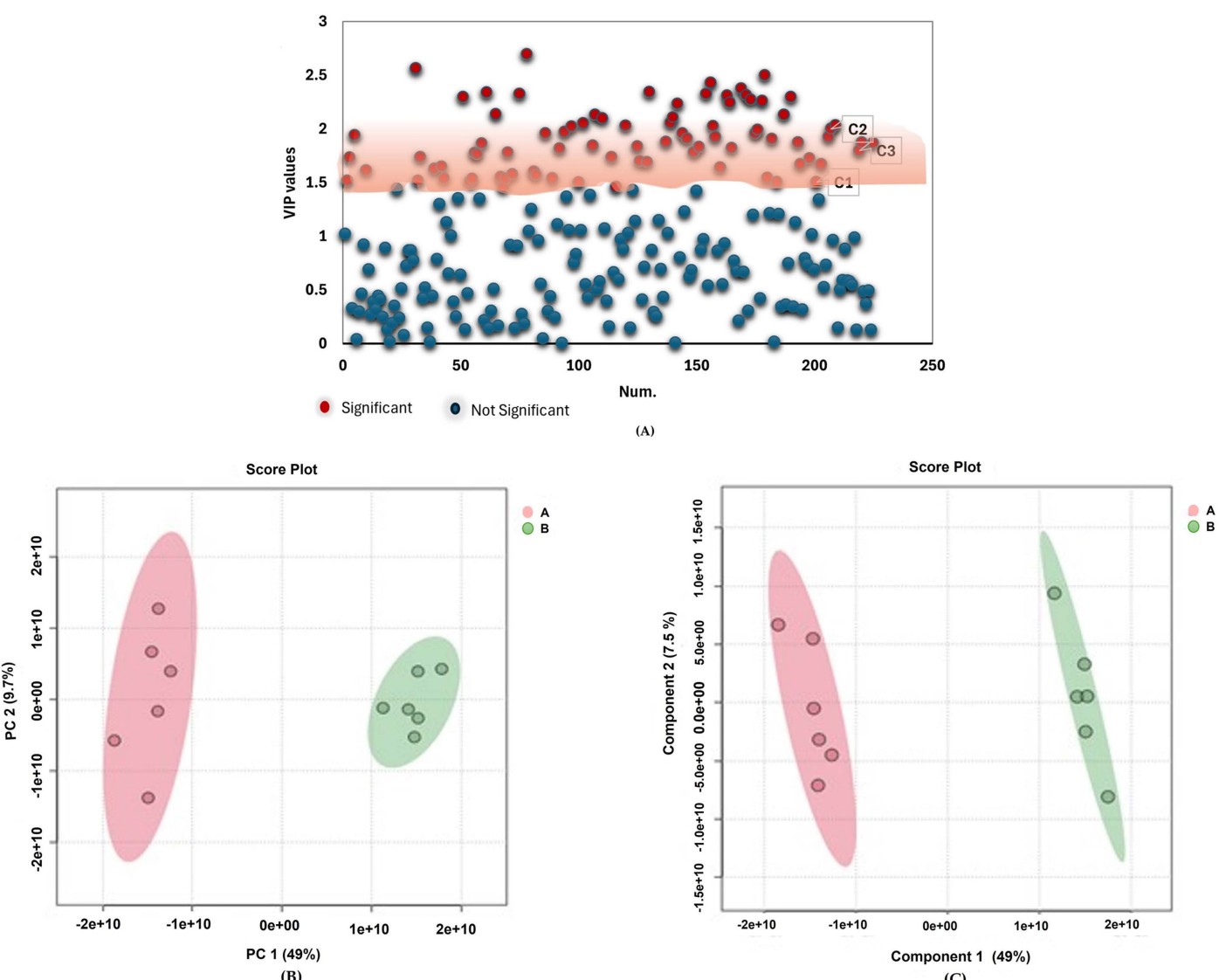

**Fig 2. Distribution patterns of *A. annua* and *A. afra* compounds.** (A) The scatter plot of VIP coordinates showing significant compounds (VIP > 1.5) and non-significant compounds (VIP < 1.5). (B) The PCA score plot depicting compound clusters and (C) The PLS-DA scores plot showing compound clusters. Key: [A]*A. annua*, [B]*A. afra,* [C1] Quercetin 3-O-(6"-acetyl-glucoside), [C2] Quercetin 3-(6"-malonyl-glucoside) and [C3] 2"-O-Acetylrutin.

fold change (FC) > 2.0, VIP > 1.5, and p value < 0.05 were chosen as signature compounds. Principal component analysis (PCA) scores plot reveals a noticeable clustering pattern between the groups of *A. annua* and *A. afra* through PC1 indicating variance of 49% (Fig 2B). The quality of a PCA model was evaluated using R2 (cum), R2X (cum) and Q2 (cum) as criteria. The metrics of 1.0 R2 (cum), 0.799 R2X (cum) and 0.697 Q2 (cum) denoted great fitness and good prediction capability of the PCA model. The findings of PCA scores plot are further supported by PLS-DA scores plot with 49% variance (Fig 2C), illustrating the metabolites that possess noteworthy compounds. The R2 denotes the great degree of fitness for PLS-DA with 0.793 R2X (cum) and 1 R2Y (cum) and 0.999 Q2 (cum) values. Hierarchical cluster heatmap consolidates these findings by displaying a grouping of compounds from *A. annua* and *A. afra* according to their concentrations (Fig 3).

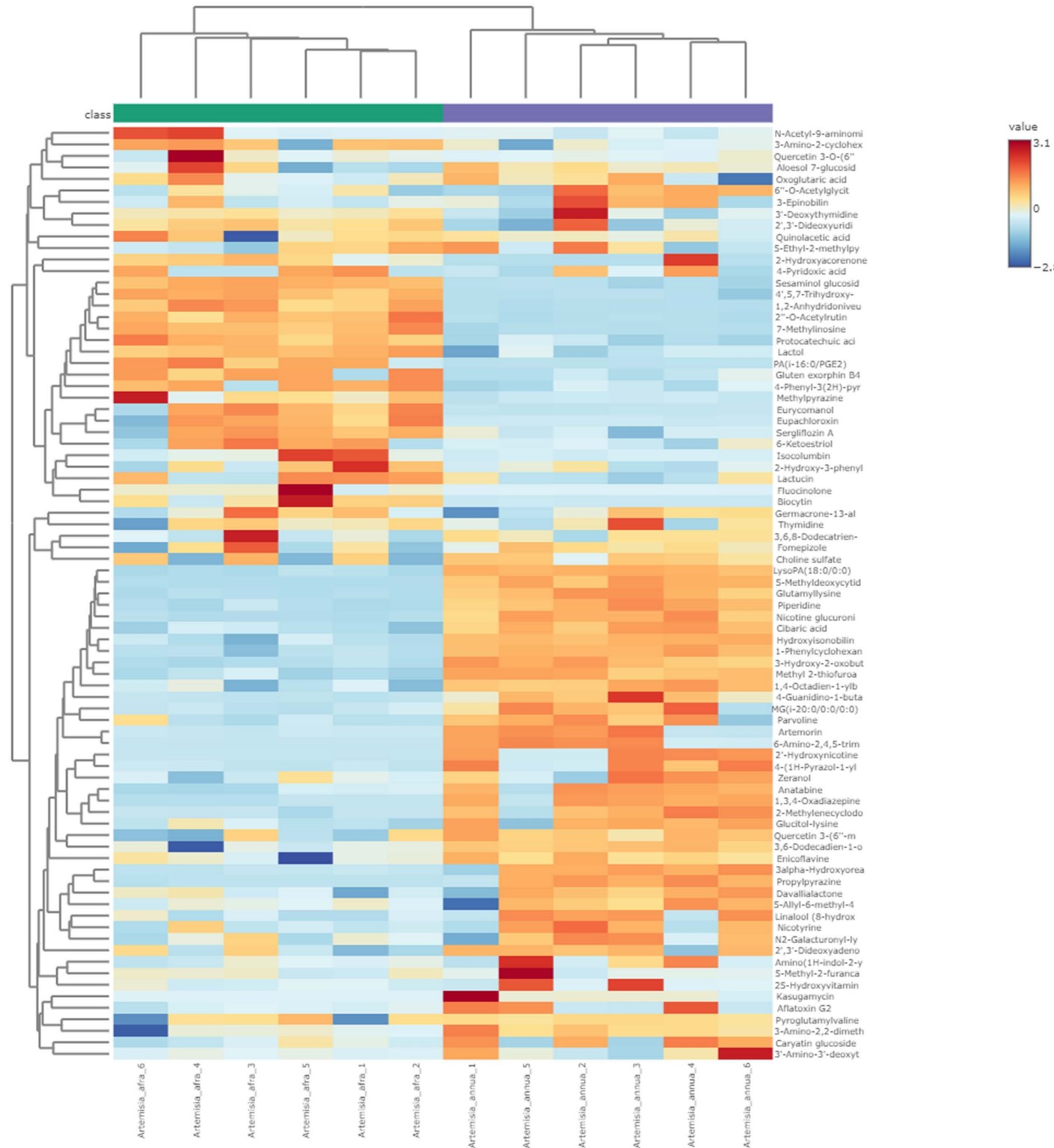

**Fig 3. Hierarchical clustering and heatmap for identification of trends in relative abundance variations of compounds from *A. annua* and *A. afra*.**

## Binding site prediction of SARS-CoV-2 3C-like protease (3CLpro)

This study utilized the PrankWeb tool to predict and visualize the protein-ligand binding sites of the SARS-CoV-2 3C-like protease (6M2N). The tool identified pockets in the 3C-like protease that strongly tends to bind to ligands, as indicated by Fig 4. This prediction is supported by the pocket score and probability score. The pocket score denotes a distinct area of the protein that functions as a site for interaction with ligands. On the other hand, the probability score is a numerical value between 0 and 1. The closer the probability of an event is to 1, the more likelihood of the predicted pocket being an actual ligand-binding site. Out of 40 pockets, pocket 1 to 6 exhibited high probability of above > 0.8. The residues forming these pockets have four chains of A, B, C and D with the average probability score of 0.923, 0.918, 0.842, 0.839, 0.795 and 0.771, respectively. This result indicates that the pocket of chain A can be predicted as an actual ligand-binding site. Moreover, pockets 1 to 6 comprised of 36, 38, 19, 20, 22 and 21 amino acids and they exhibited an average conservation score of 2.629, 2.586, 3.164, 3.105, 3.084 and 3.021, respectively. The conservation score measures the level of conservation of the amino acid residues in the projected pocket among other species, indicating that the residues in Fig 4 are strongly conserved, with Chain A exhibiting the highest probability score (0.923). Notably, Pocket 1 (Chain A) spatially overlapped with the catalytic dyad residues His41 and Cys145.

## Molecular docking

This study conducted molecular docking simulations to analyze the interactions between pro-filed metabolite compounds with different binding affinities as test compounds against SARS-CoV-2 3CLpro (PDB code: 6M2N) and were ranked based on their binding energy (kcal/mol). Among the 81 compounds (Table 1), three compounds, namely quercetin 3-O-(6"-acetyl-glucoside), 2"-O-acetylrutin, and quercetin 3-(6"-malonyl-glucoside) all bind to the active site of SARS-CoV-2 3CLpro, with the best binding affinities of -9.3 kcal/mol, -9.5 kcal/mol, and -9.3 kcal/mol, respectively. All three compounds establish numerous interactions with the active site of SARS-CoV-2 3CLpro, as indicated by Fig 5.

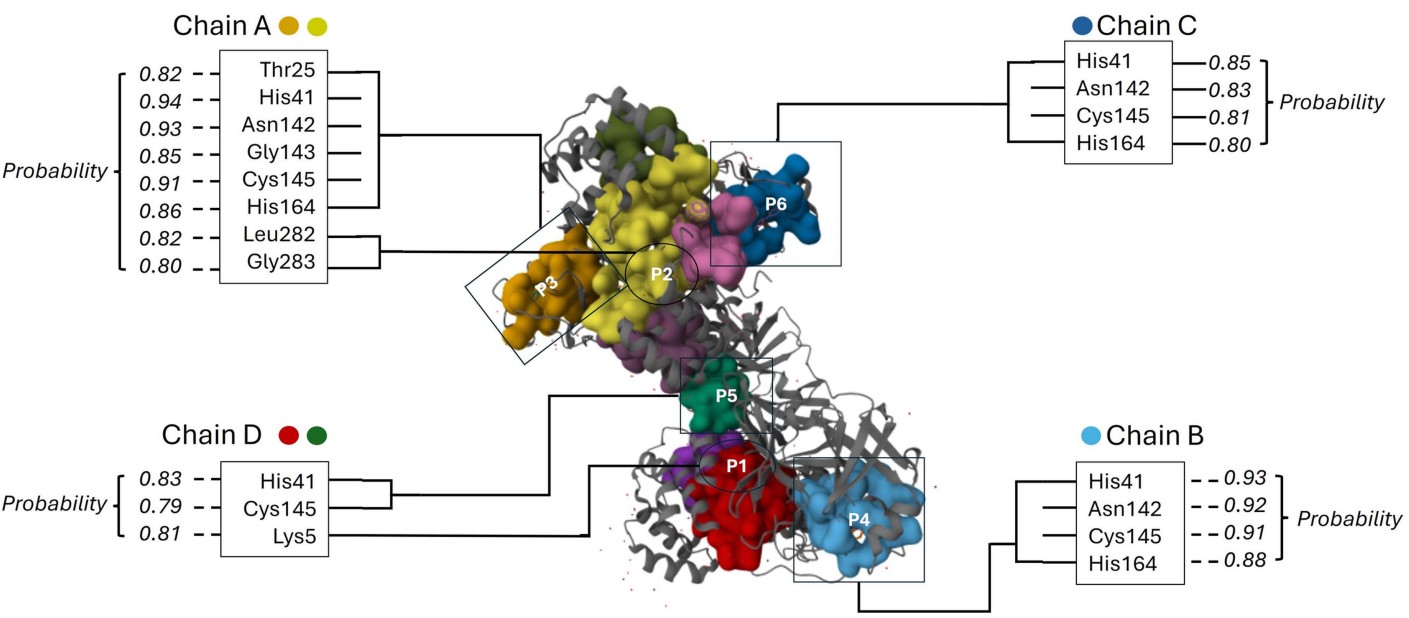

**Fig 4. The predicted binding site of the 3C-like protease (3CLpro).** The pockets visualized by amino acids residues with > 0.8 probability of binding to the site of the test compounds. Key: P1-P6 Pocket 1 to 6.

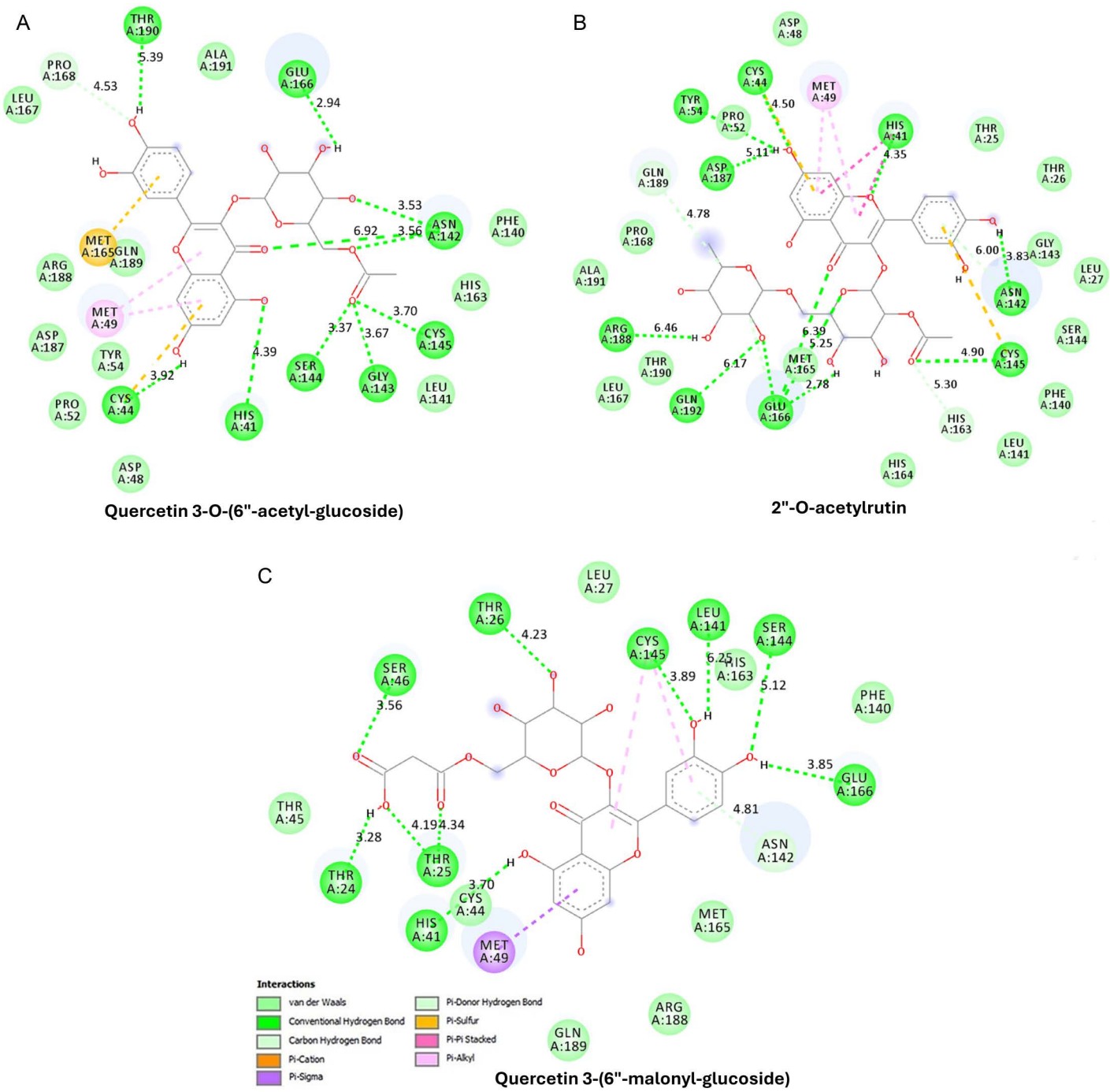

**Fig 5. Interactions of receptor-binding domain (RBD) of SARS-CoV-2 3CLpro with the three ligands of Quercetin 3-O-(6"-acetyl-glucoside), 2"-O-acetylrutin, and Quercetin 3-(6"-malonyl-glucoside).**

These interactions of receptor-binding domain (RBD) with the three ligands were enabled by various non-covalent forces which comprised of hydrogen bonds, van der Waals forces, and pi-stacking (Fig 5). As illustrated in Fig 5. the molecular interaction of Quercetin 3-O-(6"-acetyl-glucoside) with SARS-CoV-2 Mpro. The ligand interacts with multiple protein

residues through hydrogen bonding, Pi-Sulfur, Pi-Alkyl, and van der Waals forces. Hydrogen bonds are formed between the ligand's carbonyl and hydroxyl groups and key active site residues. Specifically, the ligand's carbonyl group forms three hydrogen bonds with Asn142 at distances of 3.53 Å, 3.56 Å, and 6.92 Å, and another hydrogen bond with Cys145 at 3.70 Å. Additionally, the glycosidic hydroxyl group establishes a hydrogen bond with Glu166 at 2.94 Å, while the acetyl carbonyl of the glycoside moiety interacts with Ser144 (3.37 Å) and Gly143 (3.67 Å), further stabilizing the complex. The hydroxyl group of the ligand's phenolic structure forms hydrogen bonds with His41 at 4.39 Å and Thr190 at 5.39 Å. Besides hydrogen bonding, Pi-Sulfur interactions between the ligand's aromatic ring and Met165 enhance hydrophobic stabilization. Pi-Alkyl interactions are observed between the ligand's aromatic ring and Met49 and Tyr54, which further strengthen ligand binding through hydrophobic π-π stacking interactions. Additional van der Waals interactions involve Leu167, Pro168, Ala191, Phe140, His163, Leu141, Arg188, Asp187, Asp48, and Pro52, further contributing to ligand affinity and binding stability.

As shown in Fig 5, 2''-O-acetylrutin interacts with SARS-CoV-2 Mpro through hydrogen bonding, Pi-Sulfur, Pi-Alkyl, and van der Waals forces. The ligand's acetyl carbonyl forms a hydrogen bond with Asn142 at 6.00 Å, while the hydroxyl group establishes a hydrogen bond with Glu166 at 2.78 Å. Additionally, Cys145 forms a hydrogen bond with the ligand at 4.90 Å, reinforcing the interaction within the catalytic pocket. His41 also engages in hydrogen bonding with the ligand at 4.35 Å, while Arg188 and Gln192 contribute to hydrogen bonding at distances of 6.46 Å and 6.17 Å, respectively, with Asp187 interacting at 5.11 Å. The Pi-Sulfur interaction between the ligand's aromatic ring and Met165 enhances hydrophobic binding. Pi-Alkyl interactions are detected between the ligand's aromatic ring and Met49, further stabilizing the complex. Several residues, including Leu167, Pro168, Ala191, Phe140, His163, Leu141, Arg188, Asp187, Asp48, and Pro52, display van der Waals interactions which support the ligand binding protein.

Moreover, Quercetin 3-(6''-malonyl-glucoside) interacts with SARS-CoV-2 *Mpro* via multiple stabilizing forces (Fig 5). The ligand's carbonyl group forms a hydrogen bond with Ser46 at 3.56 Å and Asn142 at 4.81 Å. Hydroxyl groups within the ligand interact with Thr24 at 3.28 Å, while Thr25 forms two hydrogen bonds at 4.19 Å and 4.34 Å. His41 engages in hydrogen bonding with the ligand's carbonyl group at 3.70 Å, reinforcing the significance of His41 in ligand stabilization. Thr26 and Glu166 also contribute to hydrogen bonding at 4.23 Å and 3.85 Å, respectively. Ser144 interacts at 5.12 Å, while Leu141 and Cys145 form hydrogen bonds at 6.25 Å and 3.89 Å, respectively. Pi-Alkyl interactions between the ligand's aromatic ring and Met49 contribute to increased complex stability through hydrophobic interactions. Additional van der Waals interactions involve Leu27, Leu141, Phe140, His41, Arg188, Gln189, and Ala191, which enhance ligand affinity and support stable binding.

## Molecular dynamics simulation

The Root Mean Square Deviation (RMSD) plot for the complexes of the three identified compounds against SARS-CoV-2 3Clpro is illustrated in Fig 6A. The Quercetin 3-O-(6''-acetyl-glucoside), 2''-O-acetylrutin, and quercetin 3-(6''-malonyl-glucoside demonstrates relative stability during the 50 ns simulations duration. The average RMSD for all complexes is 1.5 Å, which falls within the expected range (< 3 Å) for protein complexes. Moreover, the general Root Mean Square Fluctuation (RMSF) values of all selected compounds bound to SARS-CoV-2 3CLpro are observed to be lower than < 3 Å. This implies that the compounds are more rigidly bound to SARS-CoV-2 3CLpro. This also indicates that the amino acids in SARS-CoV-2 3CLpro tend to be more stable. The RMSF values of the individual residues of the bound compounds are depicted in Fig 6B. Typically, residues with the highest RMSF

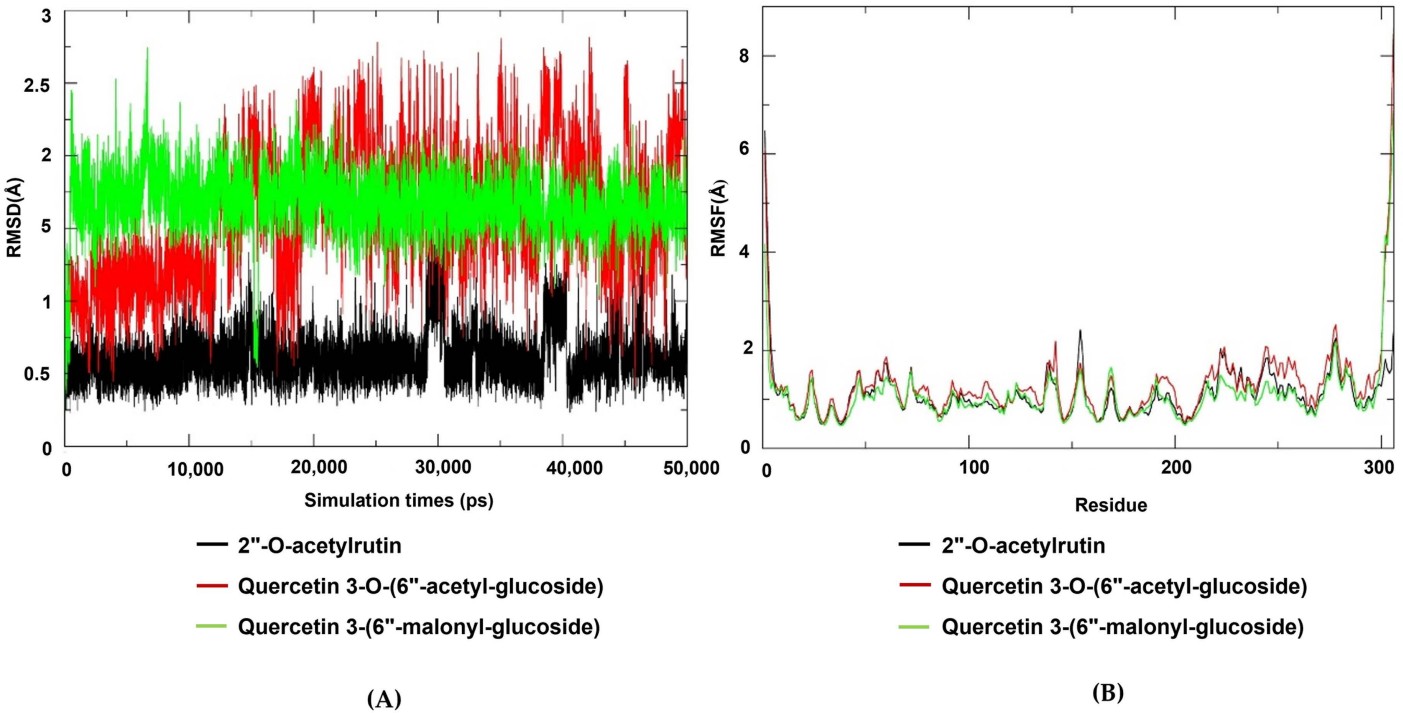

**Fig 6. Molecular dynamics simulation of quercetin 3-O-(6"-acetyl-glucoside), 2"-O-acetylrutin, and quercetin 3-(6"-malonyl-glucoside) on SARS-CoV-2 3CLpro for the period of 50 ns.** (A) Root mean square fluctuation (RMSF) plot. (B) Root mean square deviation (RMSD).

values are located in the flexible loops and regions of the proteins, while those with the lowest RMSF values are found in the more stable regions of the proteins.

Based on the molecular dynamics simulation results presented in Table 2, the analysis stability and interaction profiles of three quercetin derivatives of Quercetin 3-(6"-malonyl-glucoside), Quercetin 3-O-(6"-acetyl-glucoside), and 2"-O-acetylrutin, reveal distinct characteristics. The total binding free energy (ΔG_binding) values indicate that 2"-O-acetylrutin exhibits the most favorable interaction stability with a value of -11.92 kcal/mol, followed by Quercetin 3-(6"-malonyl-glucoside) at -6.05 kcal/mol, while Quercetin 3-O-(6"-acetyl-glucoside) shows the least stable interaction at -1.24 kcal/mol. This trend is supported by the gas phase energy (ΔG_gas) and solvation energy (ΔG_solv) components, where 2"-O-acetylrutin demonstrates more negative values, suggesting stronger intrinsic interactions and favorable solvation effects. Additionally, the van der Waals (VDWAAIS) and electrostatic (EEL) contributions further elucidate the intermolecular forces, with 2"-O-acetylrutin displaying the most significant van der Waals interactions and moderate electrostatic effects, contributing to its overall stability. These findings underscore the differential stability and interaction mechanisms among the three quercetin derivatives, highlighting the difference of functional group bound to flavonoid core influence their biological activities.

## In vitro enzyme inhibition of SARS-CoV-2 3CLpro

The enzymatic inhibitory assay was conducted to further validate the SARS-CoV-2 3CLpro inhibitory activity of the selected compounds from In silico studies. Moreover, plant extracts of *A. annua* and *A. afra* were subjected to a similar assay. Both the plant extracts and compound samples showed the ability to inhibit the enzyme activity of 3CLpro at different

**Table 2. Molecular Mechanics Poisson-Boltzmann Surface Area (MMPBSA) analysis post molecular dynamics simulation energy components for quercetin derivatives.**

| Energy Component | Quercetin 3-(6"-malonyl-glucoside) | | | Quercetin 3-O-(6"-acetyl-glucoside) | | | 2"-O-acetylrutin | | |
|---|---|---|---|---|---|---|---|---|---|
| | Average (kcal/mol) | Std. Dev. (kcal/mol) | Std. Err. of Mean (kcal/mol) | Average (kcal/mol) | Std. Dev. (kcal/mol) | Std. Err. of Mean (kcal/mol) | Average (kcal/mol) | Std. Dev. (kcal/mol) | Std. Err. of Mean (kcal/mol) |
| VDWAALS (van der Waals) | -64.43 | 4.26 | 0.19 | -44.96 | 4.42 | 0.2 | -79.17 | 4.42 | 0.2 |
| EEL (Electrostatic) | -5.99 | 6.69 | 0.3 | -120.7 | 15.92 | 0.71 | -33.55 | 8.37 | 0.37 |
| EPB (Polar Solvation) | 32.99 | 5.65 | 0.25 | 139.31 | 14.72 | 0.66 | 64.53 | 6.89 | 0.31 |
| ENPOLAR (Non-Polar SASA) | -33.59 | 1.69 | 0.08 | -24.97 | 2.01 | 0.09 | -40.14 | 1.65 | 0.07 |
| EDISPER (Non-Polar Dispersion) | 64.97 | 2.29 | 0.1 | 50.08 | 3.48 | 0.16 | 76.41 | 1.96 | 0.09 |
| $\Delta G\_gas$ (Gas Phase Energy) | -70.42 | 7.01 | 0.31 | -165.65 | 17.05 | 0.76 | -112.73 | 8.97 | 0.4 |
| $\Delta G\_solv$ (Solvation Energy) | 64.37 | 5.85 | 0.26 | 164.42 | 15.62 | 0.7 | 100.81 | 6.94 | 0.31 |
| $\Delta G\_binding$ (Total Binding Free Energy) | -6.05 | 4.99 | 0.22 | -1.24 | 6.11 | 0.27 | -11.92 | 5.7 | 0.26 |

concentrations. As presented in Fig 7, Quercetin 3-O-(6"-acetyl-glucoside), 2"-O-acetylrutin and Quercetin 3-(6"-malonyl-glucoside) inhibited around 78.94%, 89.62% and 73.67% of the enzyme activity of 3CLpro at the highest concentration of 1.5 μM, respectively. While the plant extracts of *A. annua* and *A. afra* inhibited around 82.03% and 78.86% of the enzyme activity of 3CLpro at highest concentration of 250 μg/mL, respectively (Table 3). The concentration required to inhibit 50% of the enzyme activity of 3CLpro ($IC_{50}$) was calculated. The $IC_{50}$ inhibition degree of all tested samples varied (Fig 7). All the three compounds, quercetin 3-O-(6"-acetyl-glucoside), 2"-O-acetylrutin, and quercetin 3-(6"-malonyl-glucoside) have demonstrated potent dose depended concentration on the inhibition of 3CLpro enzyme of SARS-CoV-2 with the $IC_{50}$ values of 0.12 μM, 0.10 μM and 0.17 μM, respectively. Moreover, 2"-O-acetylrutin compound exhibited a stronger inhibition effect compared to compounds of quercetin 3-O-(6"-acetyl-glucoside) and quercetin 3-(6"-malonyl-glucoside). The $IC_{50}$ value for the *A. annua* sample (10.17 μg/mL) was higher than that of the *A. afra* sample (8.38 μg/mL). *A. afra* demonstrated a stronger inhibition of SARS-CoV-2 3CLpro enzyme compared to *A. annua*. The obtained findings from In vitro enzyme inhibition of SARS-CoV-2 3CLpro assay consolidate the presented In silico studies.

## Discussion

The COVID-19 pandemic has significantly impacted global health, necessitating the development of effective treatments. Despite the development of several COVID-19 vaccines and treatments, the emergence of new SARS-CoV-2 variants challenges the efficacy of existing vaccines and treatments [15,16]. In this study, we focused on the potential of secondary metabolites from *A. afra* and *A. annua* to inhibit the main protease/ 3-chymotrypsin-like protease (*Mpro*/ 3CL*mpro*) of SARS-CoV-2. There is a need to develop new plant-based products that can target SARS-Cov-2 as suggested by the World Health Organization (WHO) in 2020 [10]. Notably, plant secondary metabolites are natural sources of antiviral compounds that can be a promising option, as most of them are safer and more cost-effective than conventional drugs [35]. However, it is important to acknowledge that some plant secondary metabolites may also have toxic effects, necessitating careful evaluation of their safety and efficacy. To

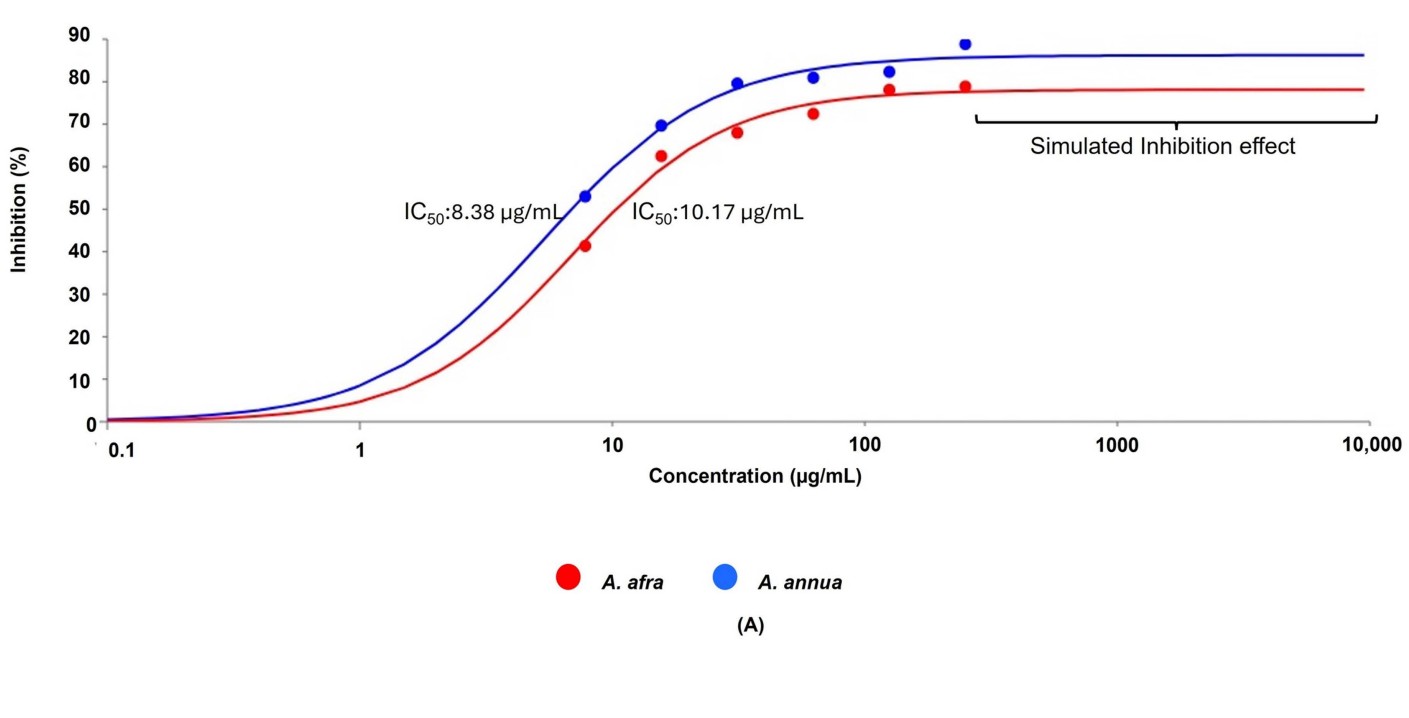

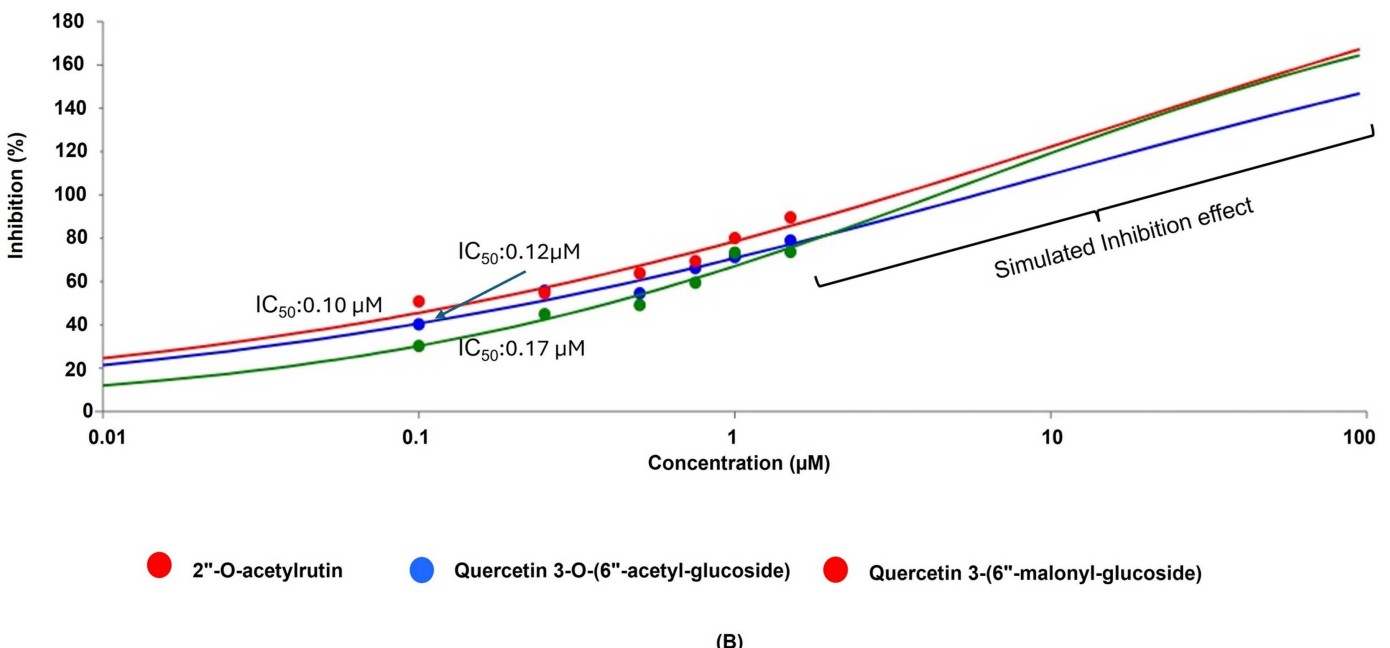

**Fig 7. Dose-dependent inhibition effect and IC50 of SARS-CoV-2 3CLpro activity exhibited by samples of (A)** *A. annua* **and** *A. afra*, **(B) Quercetin 3-O-(6"-acetyl-glucoside), 2"-O-acetylrutin, and quercetin 3-(6"-malonyl-glucoside).**

address this concern, before conducting In silico and In vitro studies, it was essential to assess the safety of the plant's leaves extracts on Vero cell lines. The selection of Vero cell lines was based on their sensitivity and reliability in revealing cytotoxic effects of plant-based products [36]. Our cytotoxicity results indicate that leaf extracts from both *A. afra* and *A. annua* did

**Table 3. SARS-CoV-2 3CLpro enzyme inhibition effect of *A. annua* and *A. afra* samples and associated compounds.**

| SARS-CoV-2 3CLpro enzymatic activity | | | | | | |
|---|---|---|---|---|---|---|
| Inhibition % of Plant samples Mean ± SE (n = 6) | | | | | | |
| *Concentrations of Plant samples* | *7.81 μg/mL* | *15.63 μg/mL* | *31.25 μg/mL* | *62.5 μg/mL* | *125 μg/mL* | *250/mL* |
| **A. afra** | 53.02 ± 0.55 | 69.67 ± 0.79 | 79.59 ± 0.48 | 80.94 ± 0.69 | 82.32 ± 0.47 | 88.86 ± 0.16 |
| **A. annua** | 41.36 ± 0.56 | 62.5 ± 0.84 | 68.01 ± 0.35 | 72.44 ± 0.25 | 78.1 ± 0.85 | 78.86 ± 0.65 |
| Inhibition % of Selected Compound samples Mean ± SE (n = 6) | | | | | | |
| *Concentrations of Compound samples* | *0.1 μM* | *0.25 μM* | *0.5 μM* | *0.75 μM* | *1 μM* | *1.5 μM* |
| **Quercetin 3-O-(6"-acetyl-glucoside)** | 40.25 ± 0.56 | 55.74 ± 0.77 | 54.54 ± 0.92 | 66.22 ± 0.79 | 71.31 ± 0.22 | 78.94 ± 0.33 |
| **2"-O-acetylrutin** | 50.82 ± 0.26 | 54.61 ± 0.58 | 63.84 ± 0.21 | 69.4 ± 0.50 | 80.06 ± 0.48 | 89.62 ± 0.43 |
| **Quercetin 3-(6"-malonyl-glucoside)** | 30.25 ± 0.26 | 44.92 ± 0.67 | 49.14 ± 0.37 | 59.47 ± 0.84 | 73.31 ± 0.21 | 73.67 ± 0.49 |

not exhibit toxic effects on Vero cells, as determined by the resazurin cell viability assay. The achieved results were concentration-dependent, decreased with increasing concentrations of *A. afra* and *A. annua* extracts exposed to Vero cell lines.

Furthermore, metabolomic profiling of the leaf extracts of South African-based *A. afra* and *A. annua* revealed different classes of secondary metabolites as triterpenoids, flavonoids, iridoid glycosides, hydroxy fatty acids, phenolic acids, carboxylic acid, phenylethanoid, and glycosides. These different classes have also been revealed from the mentioned plants [37,38], however to our knowledge, their associated potential bioactive compounds against SARS-CoV-2 3CLpro have not been explored. It can be noted that untargeted metabolomics can reveal unexpected synthetic compounds that might not have been previously known to exist in a plant, as indicated in this study. These unexpected results may provide a starting point for further investigation in terms of limited and unstandardized databases used for the diverse plant secondary metabolites identification [39]. Nonetheless, in this study, statistical and chemometric analyses revealed elucidated 81 possible signature compounds that may be bioactive of *A. annua* and *A. afra*. All revealed signature compounds were subjected to In silico studies prior to In vitro enzyme inhibition analysis, this was prompted by the fact that In silico studies significantly aid in drug discovery by predicting therapeutic potential in large scale for short periods of time, reduce cost, and enhance the likelihood of finding effective drugs when combined with other methods [40,41]. We further conducted predictions of the binding affinity of the drug candidates to target receptors, providing insight into their potential efficacy.

However, PrankWeb tool was employed first to forecast and visualize the protein-ligand binding sites of the SARS-CoV-2 3C-like protease. PrankWeb stands out as the best alternative for pocket identification when compared to other tools for various reasons such as its utilization of powerful machine learning algorithms, its user-friendly interface, its sequence conservation analysis, and its high-throughput capabilities [42,43]. This analysis revealed a specific pocket in the 3C-like protease that exhibits a strong inclination to bind with ligands. Pocket 1 (Chain A) exhibited the highest probability score and spatially overlapped with the catalytic dyad residues His41 and Cys145. This alignment demonstrates that the machine learning approach did not exclude the native ligand's binding site but instead validated its druggability through an independent, template-free methodology. These pockets accommodate various parts of the substrate and are key regions for inhibitor binding which is solvent-exposed, can interact with flexible N-terminal groups of inhibitors [44]. This aligns with our findings, as P2Rank successfully identified these regions as high-probability binding sites, further

validating its accuracy in predicting biologically relevant pockets. Importantly, P2Rank was able to accurately identify the active site binding pockets of SARS-CoV-2 3CLpro, which are located at residues His41 and Cys145. These residues have been previously identified as critical for ligand binding and catalytic activity [45,46]. The identification of these residues by P2Rank underscores its reliability in pinpointing druggable regions, even in the absence of a standard drug bound to the protein. This makes P2Rank particularly suitable for novel or less-studied targets like SARS-CoV-2 3CLpro.

Among the eighty-one screened compounds, the three best potential compounds, quercetin 3-O-(6"-acetyl-glucoside), 2"-O-acetylrutin, and quercetin 3-(6"-malonyl-glucoside) bind to the active site of SARS-CoV-2 3CLpro with the strong binding affinities of -9.3 kcal/mol, -9.5 kcal/mol, and -9.3 kcal/mol, respectively. This means that all the identified compounds are quercetin derivatives. According to a study by Banik and Bhattacharjee [47], polyphenolic chemicals like the quercetin molecule are among the most promising secondary metabolites. The most essential interactions are the hydrogen bonds established between the hydroxyl groups of the compounds and the amino acids His41 and Cys145, as they are anticipated to contribute to the suppression of 3CLpro activity. The molecule with the highest binding energy is 2"-O-acetylrutin, with a binding energy of -9.5kcal/mol. The 2"-O-acetyl group of the rutin moiety produces a hydrogen bond with the amide nitrogen of His41. The quercetin moiety further establishes a hydrogen connection with the side chain carboxylate of Cys145. The interactions of all three ligands with His41 and Cys145 are particularly significant due to their role in SARS-CoV-2 *Mpro*'s catalytic activity [46]. His41 and Cys145 form the catalytic Dyad essential for enzymatic function, where His41 acts as a general base to activate Cys145 for nucleophilic attack on the substrate [46,48,49]. Additionally, Cys145 serves as a primary target for many inhibitors, as it forms covalent bonds with reactive groups to block protease activity. His41 stabilizes inhibitor interactions, ensuring effective ligand binding and inhibition of viral replication. All three ligands demonstrated strong binding interactions with His41 and Cys145, suggesting their potential as effective inhibitors in antiviral drug development. The hydrogen bonds between the compounds and these amino acids could disrupt the catalytic mechanism of 3CLpro, leading to the inhibition of the enzyme [46–49]. These results confirm the work of Bhattacharya and co-authors [50], that revealed two other quercetin derivatives, namely, Quercetin-3-(rhamnosyl-(1->2)-alpha-L-arabinopyranoside) and Quercetin-3-neohesperidoside-7rhamnoside, with binding affinities of -8.5kcal/mol and -8.8kcal/mol, indicating a high likelihood of effectively blocking the 3CLpro enzyme activity and subsequently being promising options for future investigations and advancement as antiviral drugs targeting COVID-19.

Moreover, the stability and rigidity of a complex increases the likelihood that the inhibitor will prevent the protease from cleaving its target protein [51]. In this context, the molecular dynamics simulations conducted in this study suggest that the compounds bind more steadily to the 3CLpro-inhibitor. This stability is likely due to the relatively small and well-defined active site of the 3CLpro, which makes it more challenging for the inhibitor to diffuse away from its binding site. These findings align with previous research which found similar average RMSD values of 1.5 Å for the 3CLpro-inhibitor [52]. Furthermore, the compounds appear to be more rigidly bound to the 3CLpro molecule. This rigidity aligns with previous studies indicating that 3CLpro is a more rigid protein as compared to other targets including S glycoprotein-ligand complex [53–55]. In the case of SARS-CoV-2, the 3CLpro protease is essential for viral replication [56]. Therefore, the development of inhibitors that can target the 3CLpro protease with high affinity and stability is a promising strategy for developing antiviral therapies for SARS-CoV-2 [53].

Furthermore, previous studies have highlighted the antiviral potential of quercetin derivatives against key SARS-CoV-2 proteins. Quercetin 3-O-arabinoside 7-O-rhamnoside exhibited

strong binding affinity with papain-like protease, while quercetin 3-[rhamnosyl-(1→2)-α-L-arabinopyranoside] and quercetin-3-neohesperidoside-7-rhamnoside showed potent interactions with the spike protein receptor-binding domain and 3C-like protease, respectively [50]. These findings reinforce the high likelihood of quercetin derivatives effectively blocking SARS-CoV-2 replication and align with our results that suggest quercetin-based inhibitors, particularly 2"-O-acetylrutin, as promising antiviral drug candidates.

To further optimize these compounds for drug development, several structural modifications can be considered. First, modifications at the C-3, C-7, and C-4' positions with hydroxyl, methoxy, or fluorine groups may strengthen hydrogen bonding interactions, leading to greater stability in the active site. Additionally, adding alkyl or aryl groups could increase lipophilicity, improving membrane permeability and drug absorption. Since 2"-O-acetylrutin exhibited the strongest inhibitory activity, modifications such as replacing the acetyl group with a hydrolysis-resistant moiety (e.g., ether or carbamate) could improve metabolic stability and extend its biological half-life. Moreover, electron-donating groups at specific positions may enhance π-π interactions within the binding pocket, increasing ligand stability. Because quercetin derivatives suffer from poor water solubility, prodrug strategies such as phosphate ester derivatives or phospholipid-based formulations (e.g., Quercetin Phytosome®) have demonstrated a 20-fold increase in bioavailability, making them promising candidates for clinical use [57].

Although these In silico results provide strong evidence of quercetin derivatives as effective inhibitors, In vitro studies were essential to confirm their antiviral potential. Our findings indicate that all three compounds exhibited potent inhibitory effects on the 3CLpro enzyme of SARS-CoV-2, with 2"-O-acetylrutin demonstrating the strongest activity. Compared to 3-O-(6"-acetyl-glucoside) and quercetin 3-(6"-malonyl-glucoside), 2"-O-acetylrutin exhibited superior inhibition In vitro, aligning with its highest binding energy and lowest RMSD values. These results emphasize the therapeutic potential of 2"-O-acetylrutin as a promising candidate for further drug development against SARS-CoV-2. As revealed in both In silico and In vitro studies, 2"-O-acetylrutin demonstrates to be a potent inhibitor of SARS-CoV-2 3CLpro In vitro when compared to 3-O-(6"-acetyl-glucoside), and quercetin 3-(6"-malonyl-glucoside). This compound also exhibited the lowest RMSD with the strongest binding affinities, emphasizing its potential as effective therapeutics against SARS-CoV-2. By addressing these aspects, quercetin-based inhibitors could serve as promising antiviral therapeutics, not only for SARS-CoV-2 but also for other viral proteases of emerging coronaviruses.

## Conclusions

This study opens several avenues for future research, particularly in broadening the scope to additional plant-derived bioactive compounds and alternative viral targets. While our study focused on quercetin derivatives from selected plant sources, future investigations will explore structurally related flavonoids and polyphenolic compounds from other medicinal plants known for their antiviral properties. Promising candidates such as kaempferol derivatives, epigallocatechin gallate (EGCG), and myricetin analogs have shown significant potential against viral proteases and could be screened for their inhibitory effects on SARS-CoV-2 3CLpro, papain-like protease (PLpro), and RNA-dependent RNA polymerase (RdRp). Expanding the study to cover these compounds will help identify additional lead molecules with enhanced antiviral activity, drug-like properties, and bioavailability.

Additionally, while the current study focuses on 3C-like protease (3CLpro) as the primary target, we recognize the importance of broadening the scope to other critical viral targets involved in SARS-CoV-2 replication. Future research will investigate the potential of

quercetin derivatives in inhibiting the spike protein (S), helicase (nsp13), and PLpro, which play crucial roles in viral entry and immune evasion. Targeting multiple pathways within the viral life cycle may increase the therapeutic efficacy of these compounds and reduce the likelihood of viral resistance.

## Study limitations and future approaches

We acknowledge that while the present study provides strong in silico and In vitro evidence supporting quercetin derivatives as potential SARS-CoV-2 inhibitors, several limitations should be addressed in future work. First, the computational docking and molecular dynamics (MD) simulations, although reliable, are predictive models and require further experimental validation through crystallographic studies (X-ray or cryo-EM) to confirm the exact binding conformations and interactions of the compounds within the active site of 3CLpro.

## Supporting Information

**S1 Fig. ESI in positive ionization mode depicting base peak intensity chromatograms of (A)** *A. annua* **and (B)** *A. afra* **samples.**
(DOCX)

**S1 Table. UPLC-MS/MS distinguished metabolites in the extracts of** *A. annua* **and** *A. afra*.
(DOCX)

**S2 Table. Raw data from UPLC-MS/MS of** *A. annua* **and** *A. afra* **metabolites.**
(XLSX)

## Acknowledgments

We would like to give special appreciation to the Department of Biology and Environmental Sciences, Sefako Makgatho Health Sciences University for providing resources and laboratory facilities.

## Author contributions

**Conceptualization:** Nqobile Monate Mkolo.

**Data curation:** Karabo Maselepe Makoana, Muhammad Sulaiman Zubair, Nqobile Monate Mkolo.

**Formal analysis:** Karabo Maselepe Makoana, Muhammad Sulaiman Zubair, Nqobile Monate Mkolo.

**Funding acquisition:** Nqobile Monate Mkolo.

**Investigation:** Karabo Maselepe Makoana, Clarissa Marcelle Naidoo, Muhammad Sulaiman Zubair, Mmei Cheryl Motshudi, Nqobile Monate Mkolo.

**Methodology:** Karabo Maselepe Makoana, Muhammad Sulaiman Zubair, Nqobile Monate Mkolo.

**Project administration:** Karabo Maselepe Makoana, Clarissa Marcelle Naidoo, Mmei Cheryl Motshudi, Nqobile Monate Mkolo.

**Resources:** Nqobile Monate Mkolo.

**Software:** Muhammad Sulaiman Zubair, Nqobile Monate Mkolo.

**Supervision:** Clarissa Marcelle Naidoo, Muhammad Sulaiman Zubair, Mmei Cheryl Motshudi, Nqobile Monate Mkolo.

**Validation:** Karabo Maselepe Makoana, Clarissa Marcelle Naidoo, Mmei Cheryl Motshudi, Nqobile Monate Mkolo.

**Visualization:** Karabo Maselepe Makoana.

**Writing – original draft:** Karabo Maselepe Makoana.

**Writing – review & editing:** Clarissa Marcelle Naidoo, Muhammad Sulaiman Zubair, Mmei Cheryl Motshudi, Nqobile Monate Mkolo.

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
