## [Decision Letter · Decision Letter 0]

16 Dec 2024

PONE-D-24-46570Integration of metabolomics and chemometrics with in-silico and in-vitro approaches to unravel SARS-CoV-2 inhibitors from South African plantsPLOS ONE

Dear Dr. Mkolo,

Thank you for submitting your manuscript to PLOS ONE. After careful consideration, we feel that it has merit but does not fully meet PLOS ONE’s publication criteria as it currently stands. Therefore, we invite you to submit a revised version of the manuscript that addresses the points raised during the review process.

We look forward to receiving your revised manuscript.

Kind regards,

Ahmed A. Al-Karmalawy, PhD

Academic Editor

PLOS ONE

Journal Requirements:

2. Thank you for stating the following financial disclosure: [South African Medical Research Council, grant number MKOLO24/25]. Please state what role the funders took in the study. If the funders had no role, please state: "The funders had no role in study design, data collection and analysis, decision to publish, or preparation of the manuscript." If this statement is not correct you must amend it as needed. Please include this amended Role of Funder statement in your cover letter; we will change the online submission form on your behalf.

3. Thank you for stating the following in the Acknowledgments Section of your manuscript: [This research was funded by the South African Medical Research Council, grant number MKOLO24/25” and “The APC was funded by South African Medical Research Council”] We note that you have provided funding information that is not currently declared in your Funding Statement. However, funding information should not appear in the Acknowledgments section or other areas of your manuscript. We will only publish funding information present in the Funding Statement section of the online submission form. Please remove any funding-related text from the manuscript and let us know how you would like to update your Funding Statement. Currently, your Funding Statement reads as follows: [South African Medical Research Council, grant number MKOLO24/25] Please include your amended statements within your cover letter; we will change the online submission form on your behalf.

4. We note that your Data Availability Statement is currently as follows: [All relevant data are within the manuscript and its Supporting Information files] Please confirm at this time whether or not your submission contains all raw data required to replicate the results of your study. Authors must share the “minimal data set” for their submission. PLOS defines the minimal data set to consist of the data required to replicate all study findings reported in the article, as well as related metadata and methods (https://journals.plos.org/plosone/s/data-availability#loc-minimal-data-set-definition). For example, authors should submit the following data: - The values behind the means, standard deviations and other measures reported; - The values used to build graphs; - The points extracted from images for analysis. Authors do not need to submit their entire data set if only a portion of the data was used in the reported study. If your submission does not contain these data, please either upload them as Supporting Information files or deposit them to a stable, public repository and provide us with the relevant URLs, DOIs, or accession numbers. For a list of recommended repositories, please see https://journals.plos.org/plosone/s/recommended-repositories. If there are ethical or legal restrictions on sharing a de-identified data set, please explain them in detail (e.g., data contain potentially sensitive information, data are owned by a third-party organization, etc.) and who has imposed them (e.g., an ethics committee). Please also provide contact information for a data access committee, ethics committee, or other institutional body to which data requests may be sent. If data are owned by a third party, please indicate how others may request data access.

5. Please remove your figures from within your manuscript file, leaving only the individual TIFF/EPS image files, uploaded separately. These will be automatically included in the reviewers’ PDF**.**

Reviewers' comments:

Reviewer's Responses to Questions

Comments to the Author

1. Is the manuscript technically sound, and do the data support the conclusions?

Reviewer #1: Yes

Reviewer #2: Yes

2. Has the statistical analysis been performed appropriately and rigorously? 

Reviewer #1: Yes

Reviewer #2: Yes

3. Have the authors made all data underlying the findings in their manuscript fully available?

Reviewer #1: Yes

Reviewer #2: Yes

4. Is the manuscript presented in an intelligible fashion and written in standard English?

Reviewer #1: Yes

Reviewer #2: Yes

5. Review Comments to the Author

Reviewer #1: Manuscript entitled " Integration of metabolomics and chemometrics with in-silico 2 and in-vitro approaches to unravel SARS-CoV-2 inhibitors 3 from South African plants" submitted to PLOS ONE, has been reviewed. The article aligns well with the provided criteria, demonstrating originality, methodological rigor, and sound ethical practices. Further minor edits could enhance clarity in some sections, but overall, it meets the academic standards. Only few points are required:

1) Line 170: grammatical adjustment, correcting capitalized noun that does not require it ("Protease" to "protease").

2) Adding a statement on ethical approval or clarifying if an ethics waiver was obtained would help meet community standards for ethics and research integrity.

3) While data analysis software is named, the availability of raw or processed data isn’t explicitly stated, creating ambiguity about data transparency. A simple statement on data accessibility (e.g., “Data available upon request” or “Data deposited in [repository]”) would address this.

4) in metabolomics, especially with LC-MS (Liquid Chromatography-Mass Spectrometry), compound identification is often labeled as “tentative” unless confirmed with additional analytical methods like nuclear magnetic resonance (NMR) spectroscopy or comparison to known standards. For example, Line 80: “Metabolomics Approach for Identification and quantification of Bioactive Compounds” it would be more accurate to say, “Metabolomics Approach for Tentative Identification and quantification of Bioactive Compounds”.

5) Since the study focuses on specific active compounds like quercetin derivatives, it might be beneficial to explicitly mention any efforts to cross-validate these findings with other methods or databases. For instance, mentioning if the identified compounds’ spectral data were cross-referenced with reputable databases (e.g., Dictionary of Natural Products (DNP), Metlin, HMDB) could enhance confidence in the tentative identification.

6) Several compounds identified in this study, including fomepizole, methyl 2-thiofuroate, 3-amino-2-cyclohexenone, and 3-amino-2,2-dimethylpropanoic acid, are synthetic and not known to occur naturally. Zeranol, while derived from a natural mycotoxin (zearalenone), is itself a semi-synthetic compound. This raises questions about the study’s methodology for profiling natural products. Clarification on the verification steps taken to ensure the accuracy of compound identification would strengthen the study’s conclusions, particularly in the context of natural product research.

Reviewer #2: The authors of the provided manuscript evaluated the anti-SARS-Cov-2 activity of identified South African plant metabolites through experimental and computational approaches. This manuscript is relevant, valuable in the field of drug discovery, and with potentiality for high citation. Few points are to be addressed prior publication.

1. Authors should elaborate more, providing more details on the target topology and pocket description prior presenting the docking findings.

2. Authors should provide a relevant rational for adopting the machine learning-ligand binding pocket prediction rather than the pocket depicted for the co-crystallized SARS-Cov-2 inhibitor (3WL; 5,6,7-trihydroxy-2-phenyl-4H-chromen-4-one).

3. Authors should elaborate more on the ligand-target interaction patterns Bonding should be annotated in terms of bond distances. Specially for Hydrogen bonding, this type of compound-protein polar interaction should be presented within hydrogen bond distances as well as bond angles since hydrogen bond depend on both. Authors should mention the Hydrogen bond angles as well as their distances, since the strength of hydrogen bonding is based on both parameters in a way to ensure the adequacy of optimum hydrogen bonding.

4. The author should provide dissection of the free binding energies MM-GBSA or MM-PBSA calculations estimated from the MD simulation trajectories. This would obtain total energy calculations as well as different energy contribution terms (∆G electrostatic, ∆G van der Waal, ∆G polar solvation, ∆G non-polar SASA solvation,…) to identify the dominant energy terms that can guide further ligand optimization and development.

Further, the total free binding energies for compound-target complex should be broken down to the target residue energy contribution to highlight which residues impose the great impact on ligand binding for future targeting and lead optimization

5. Authors are advised to provide snap shoot of the MD simulated ligand-protein complex at different time intervals (example 0ns, 20ns, 40ns, and/or 50ns) to track the main conformational/orientation changes for both the compound and surrounding residues over time.

6. Based on the study results, what are the take-away messages. Authors are advised to highlight the suggested structural modifications that would improve the compound’s biological activities based on the in silico findings. These insights would be beneficial for guiding further lead optimization and development.

7. Finally, concerning the conclusion, authors are advised to elaborate more on the future of this work? Will you broaden the scope to another plant source and/or drug target? What are the study limitations and what approaches could be conducted to further address them?

6. PLOS authors have the option to publish the peer review history of their article (what does this mean? ). If published, this will include your full peer review and any attached files.

**Do you want your identity to be public for this peer review?** For information about this choice, including consent withdrawal, please see our Privacy Policy .

Reviewer #1: No

Reviewer #2: **Yes**

---

## [Author Response · Author response to Decision Letter 1]

10 Feb 2025

Kindly find the::

A rebuttal letter that responds to each point raised by the academic editor and reviewer(s). A separate file labeled 'Response to Reviewers' was uploaded.

---

## [Decision Letter · Decision Letter 1]

19 Feb 2025

Integration of Metabolomics and Chemometrics with In-Silico and In-Vitro Approaches to Unravel SARS-Cov-2 Inhibitors from South African Plants

PONE-D-24-46570R1

Dear Dr. Mkolo,

We’re pleased to inform you that your manuscript has been judged scientifically suitable for publication and will be formally accepted for publication once it meets all outstanding technical requirements.

Kind regards,

Ahmed A. Al-Karmalawy, PhD

Academic Editor

PLOS ONE

Reviewers' comments:

Reviewer's Responses to Questions

**Comments to the Author**

1. If the authors have adequately addressed your comments raised in a previous round of review and you feel that this manuscript is now acceptable for publication, you may indicate that here to bypass the “Comments to the Author” section, enter your conflict of interest statement in the “Confidential to Editor” section, and submit your "Accept" recommendation.

Reviewer #2: (No Response)

2. Is the manuscript technically sound, and do the data support the conclusions?

Reviewer #2: (No Response)

3. Has the statistical analysis been performed appropriately and rigorously? 

Reviewer #2: (No Response)

4. Have the authors made all data underlying the findings in their manuscript fully available?

Reviewer #2: (No Response)

5. Is the manuscript presented in an intelligible fashion and written in standard English?

Reviewer #2: (No Response)

6. Review Comments to the Author

Reviewer #2: (No Response)

7. PLOS authors have the option to publish the peer review history of their article (what does this mean? ). If published, this will include your full peer review and any attached files.

**Do you want your identity to be public for this peer review?** For information about this choice, including consent withdrawal, please see our Privacy Policy .

Reviewer #2: **Yes**

---

## [Editor Report · Acceptance letter]

PONE-D-24-46570R1

PLOS ONE

Dear Dr. Mkolo,

I'm pleased to inform you that your manuscript has been deemed suitable for publication in PLOS ONE. Congratulations! Your manuscript is now being handed over to our production team.

Kind regards,

on behalf of

Associate Professor Ahmed A. Al-Karmalawy

Academic Editor

PLOS ONE